# Control of Golgi- V-ATPase through Sac1-dependent co-regulation of PI(4)P and cholesterol

Xin Zhou [1,2,8], Miesje M. van der Stoel [1,2,8], Shreyas Kaptan[3], Haoran Li[1,2], Shiqian Li[1,2], Maarit Hölttä [1,2], Helena Vihinen [4], Eija Jokitalo [4], Christoph Thiele [5], Olli Pietiläinen [6], Shin Morioka [7], Junko Sasaki[7], Takehiko Sasaki [7], Ilpo Vattulainen [3] & Elina Ikonen [1,2] ✉

Sac1 is a conserved phosphoinositide phosphatase, whose loss-of-function compromises cell and organism viability. Here, we employ acute auxin-inducible Sac1 degradation to identify its immediate downstream effectors in human cells. Most of Sac1 is degraded in ~1 h, paralleled by increased PI(4)P and decreased cholesterol in the *trans*-Golgi network (TGN) during the following hour, and superseded by Golgi fragmentation, impaired glycosylation, and selective degradation of TGN proteins by ~4 h. The TGN disintegration results from its acute deacidification caused by disassembly of the Golgi V-ATPase. Mechanistically, Sac1 mediated TGN membrane composition maintains an assembly-promoting conformation of the $V_0a2$ subunit. Key phenotypes of acute Sac1 degradation are recapitulated in human differentiated trophoblasts, causing processing defects of chorionic gonadotropin, in line with loss-of-function intolerance of the human *SACM1L* gene. Collectively, our findings reveal that the assembly of the Golgi V-ATPase is controlled by the TGN membrane via Sac1 fuelled lipid exchange.

PI(4)P is the most abundant species of phosphoinositides that serve as major regulators of organelle identity and dynamics[1]. It is asymmetrically distributed in the cell, with high levels in the *trans*-Golgi network (TGN) and the plasma membrane (PM)[2], regulated by an interplay between organelle specific PI4-kinases (PI4Ks) and the endoplasmic reticulum (ER)/Golgi integral phosphatase Sac1[3–5]. In the Golgi, PI(4)P plays an important role by controlling the localization, signaling and activity of PI(4)P binding proteins and subsequently lipid homeostasis and vesicular trafficking[6–11]. Defects in the regulation of cellular PI(4)P levels are linked to neurodegenerative diseases[12] and cancer[13,14].

Importantly, PI(4)P in the TGN serves as counter-exchange cargo for lipid transfer reactions at membrane contact sites. Here, PI(4)P regulates the docking of lipid transfer proteins, such as oxysterol-binding protein (OSBP), which transfers PI(4)P from the TGN to the ER and in exchange relocates cholesterol to the opposite direction, from the ER to the TGN[15–17]. At the ER, Sac1 hydrolyzes PI(4)P into phosphatidylinositol (PI), thereby creating an intracellular PI(4)P gradient that drives anterograde cholesterol transport and reduces PI(4)P levels in the TGN[16,18,19]. Besides OSBP, other lipid transporters such as ORP9 and Asters/GRAMDs, are also involved in maintaining the cholesterol distribution between the ER and TGN[20]. Sac1 depletion causes an increase in PI(4)P[10,21] and results in enlargement of the Golgi, mislocalization of glycosylation enzymes and glycosylation defects[22,23]. Additionally, by locally controlling Golgi PI(4)P levels, Sac1 can

[1]Stem Cells and Metabolism Research Program and Department of Anatomy, Faculty of Medicine, University of Helsinki, Helsinki, Finland. [2]Minerva Foundation Institute for Medical Research, Helsinki, Finland. [3]Department of Physics, University of Helsinki, Helsinki, Finland. [4]Institute of Biotechnology, University of Helsinki, Helsinki, Finland. [5]Life and Medical Science Institute, University of Bonn, Bonn, Germany. [6]Neuroscience Center, Helsinki Institute of Life Science, University of Helsinki, Helsinki, Finland. [7]Medical Research Institute, Tokyo Medical and Dental University, Tokyo, Japan. [8]These authors contributed equally: Xin Zhou, Miesje M. van der Stoel. ✉e-mail: elina.ikonen@helsinki.fi

regulate cargo trafficking and secretion[23,24]. However, how Sac1 mediated cholesterol and PI(4)P distribution affects Golgi functioning and which critical effectors are involved in this, are currently unknown.

Sac1, or suppressor of actin-1, was first discovered in yeast, where it regulates actin cytoskeletal organization and Golgi secretion[25,26]. Sac1 is the only known PI(4)P phosphatase in mammalian cells and its catalytic domain is highly conserved across species[27]. Constitutive loss of Sac1 compromises the viability of cells and organisms, causing early embryonic lethality in e.g. *Drosophila* and mice[22,28]. As long-term loss of Sac1 compromises viability, the precise mechanisms of how Sac1 fuelled lipid exchange affects the function of the Golgi have been challenging to study. Indeed, due to the rapid turnover and broad distribution of phosphoinositide species in cellular membranes, most if not all membrane compartments may become afflicted in chronic Sac1 depletion models.

Here, we employed auxin-inducible degradation (AID) of Sac1 to rapidly deplete the protein in several human cell types, allowing us to investigate early effects after Sac1 loss. We identified vacuolar-type ATPase (V-ATPase) as a key effector of Sac1 controlled PI(4)P and cholesterol content in the TGN membrane, impacting TGN pH and resulting in defects in Golgi terminal glycosylation and secretion, with physiological implications for early embryonic development. Mechanistically, increased PI(4)P levels in the TGN characteristic of Sac1 loss, promoted a conformational change of the $V_0a2$ subunit of the Golgi V-ATPase, preventing the assembly of the V-ATPase holocomplex. Taken together, this study reveals the Golgi V-ATPase as a critical effector of Sac1 fuelled PI(4)P/cholesterol exchange to regulate Golgi integrity and cargo processing.

## Results

### Sac1 exerts acute control of Golgi morphology

To investigate the acute consequences of Sac1 depletion on the Golgi, we took advantage of the AID system that we recently established in several human cell types[29,30] (Fig. 1a). Within 1 h after IAA addition, about 90% of endogenous Sac1 was removed from A431 cells (Fig. 1b). At this time, cellular PI(4)P was already moderately increased, as assessed by mass spectrometry[31], and after 2 h of IAA treatment, the PI(4)P content was robustly elevated, to ~3-fold from the starting level (Fig. 1c). Instead, no significant or very minor changes were observed in PI(3)P, PI(4,5)$P_2$, and PI(3,5)$P_2$ species at this time (Supplementary Fig. 1a–c), highlighting PI(4)P as the primary substrate of Sac1 in cells. All acyl chain compositions of PI(4)P species were similarly affected, whereas no changes in the acyl chain composition of the other PIP species were observed (Supplementary Fig. 1d). By 4 h of Sac1 depletion, all detected phosphoinositide species were elevated compared to the starting situation (Supplementary Fig. 1e–h).

Previously, Sac1 depletion was shown to affect Golgi morphology[10,22–24]. To investigate the effect of Sac1 degradation on Golgi integrity, we performed immunofluorescence staining using Golgi markers. After 2 h of IAA induction, the normally tight perinuclear clustering of *cis*- and *trans*-Golgi elements (GM130 and TGN46 as markers, respectively) was lost and the ratio of cells with aberrantly fragmented Golgi increased (Fig. 1d, e). By 4 h, GM130-positive structures were largely fragmented and TGN46 was hardly detectable by fluorescence microscopy (Fig. 1d, e). This was reflected by reduced TGN46 protein levels, while the protein levels of Golgin-97 (*trans*-), GRASP55 (medial-) and GM130 (*cis*-Golgi) remained unchanged (Fig. 1f).

Consistent with these results, thin-section transmission electron microscopy (TEM) revealed a rapid transformation of Golgi ribbons to tubular-vesicular networks in 2 h of IAA induction (Fig. 1g). While the Golgi had a typical ribbon-like organization in 80% of the DMSO treated cells, the Golgi elements were dispersed in 70% of the Sac1 degraded cells: the occurrence and length of stacks decreased, and the Golgi elements were mainly compromised of tubular networks and tubulo-vesicular clusters (Fig. 1g and Supplementary Data 1).

Importantly, the rapid fragmentation of the Golgi upon Sac1 removal was also evident in other Sac1-degron models, such as HEK293 and A549 cells (Supplementary Fig. 1i, j). These results show that Sac1 is important to maintain Golgi integrity.

### Sac1 phosphatase activity is required for terminal glycosylation

To obtain an unbiased overview of the effects of acute Sac1 removal at the protein level, we performed quantitative proteomic analysis of A431 Sac1-degron cells. After 3 h of IAA induction, a total of 8040 proteins were identified, of which 203 were upregulated (fold change ≥ 1.3, IAA to DMSO) and 103 downregulated (fold change ≤ 0.77, IAA to DMSO) (Supplementary Fig. 2a, Supplementary Data 2, Supplementary Fig. 3). Among the most upregulated proteins in the Sac1-degron cell proteomics were PI(4)P binding proteins, such as OSBP, ORP9 and CERT1 (Supplementary Fig. 2a). Gene ontology (GO) analysis revealed that the most downregulated proteins were enriched in the biological processes and cellular components related to glycosylation, Golgi organization and membrane (Fig. 2a and Supplementary Data 2), and the most upregulated proteins were enriched in membrane trafficking and organelle organization (Supplementary Fig. 2b and Supplementary Data 2). Interestingly, while the levels of ER and *cis*-/medial-Golgi glycosylation enzymes were unaltered or moderately downregulated, the enzymes of the *trans*-Golgi cisternae, in particular galactosyl- and sialyltransferases, were strongly downregulated (Fig. 2b).

Western blotting confirmed the robust reduction of beta-1,4-galactosyltransferase 1 (B4GALT1) and of TGN46 (Fig. 2c, d) by 4 h of IAA induction, while the mRNA levels of B4GALT1 and TGN46 remained unchanged (Supplementary Fig. 2c). To investigate if lysosomal degradation plays a role in the loss of TGN46, we neutralized lysosomes using Bafilomycin A1 (BafA1) during IAA treatment. This effectively restored TGN46 protein levels (Supplementary Fig. 2d), suggesting that TGN46 undergoes lysosomal proteolysis upon Sac1 depletion. Interestingly, at 8 h the mature form of TGN46 was replaced by an immature form of the protein (Supplementary Fig. 2e), and PNGaseF and α−2-3,6,8,9-Neuraminidase A treatments indicated defects in its terminal glycosylation (Supplementary Fig. 2f, g). Of note, 4 h treatment of OSW-1 also reduced TGN46 protein levels (Supplementary Fig. 2h, i), indicating that OSBP inhibition phenocopied acute Sac1 depletion.

Sac1-dependent functions rely on its catalytic activity[18,32]. To scrutinize if the effect of Sac1 on TGN integrity was dependent on its phosphatase activity, we reintroduced EGFP-tagged wild-type Sac1 (Sac1[wt]) and catalytically inactive mutant Sac1-C389S (Sac1[mut]) to Sac1 degraded cells, using the well transfectable HEK293A Sac1-degron cells. After 24 h of transient transfection, endogenously degron-tagged Sac1 was removed by 6 h of IAA induction, while the levels of exogenous Sac1[wt] and Sac1[mut] remained stable (Fig. 2e). In this setting, the reduction of B4GALT1 levels caused by Sac1 degradation could be rescued by reintroduction of Sac1[wt] but not by Sac1[mut] (Fig. 2e, f). Collectively, these results suggest that Sac1 mediated PI(4)P hydrolysis maintains TGN integrity and terminal glycosylation.

### Sac1 is needed to maintain TGN pH and Golgi V-ATPase assembly status

The Golgi maintains a proton gradient, from neutral pH at the *cis*-face to pH ~6 at the TGN. This is important to maintain Golgi integrity and protein glycosylation[33]. To assess if acute Sac1 degradation affects the pH of the TGN, we transfected Sac1-degron cells with TGN-targeting GalT-RpHLuorin2, a pH-sensitive GFP mutant fused to B4GALT1[34] (Supplementary Fig. 4a). Fluorescence lifetime values of GalT-RpHLuorin2 were obtained from A431 Sac1-degron cells and converted to pH values according to a standard curve (Supplementary Fig. 4b).

The average pH of the TGN of A431 cells remained consistently ~5.9 in the DMSO control group over time, while it was elevated to ~6.1 at 2 h and ~6.3 at 3 h of IAA induction (Fig. 3a and Supplementary

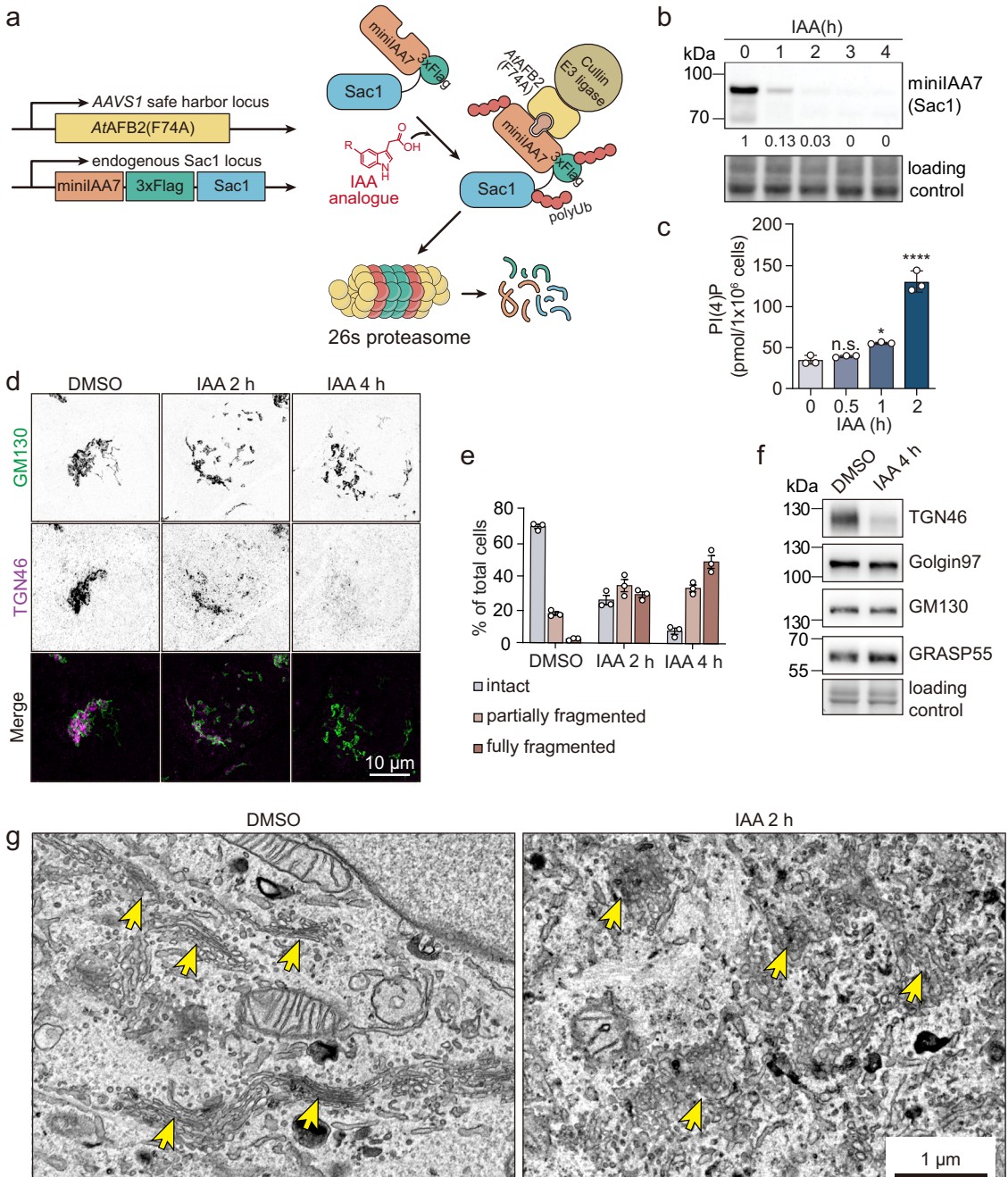

**Fig. 1 | Rapid Sac1 depletion disturbs Golgi morphology. a** Schematic overview of the genetic manipulations for auxin-inducible degradation of Sac1. Sac1 is endogenously tagged with a miniIAA7-3xFlag tag and the auxin receptor *At*AFB2(F74A) mutant is exogenously expressed in cells. After addition of an IAA analogue, miniIAA7-3xFlag-Sac1 is recruited to the AtAFB2(F74A)-Cullin E3 ligase complex, resulting in polyubiquitination and proteasomal degradation of Sac1. **b** Western blot analysis of Sac1-degron A431 cell lysates after IAA addition for 0 to 4 h in A431 cells, blotted for miniIAA7 to detect Sac1. Normalized relative intensity of each band is marked under the blot. **c** Bar graph showing total PI(4)P levels ±S.E.M. (pmol/x10⁶ cells) of Sac1-degron A431 cells treated with DMSO for 2 h (0) or IAA for 0.5, 1 or 2 h as measured by PRMC-MS. n = 3 biological replicates. One-way ANOVA with Dunnett's multiple comparisons test. n.s. p = 0.7464, * p = 0.0167, **** p < 0.0001. **d** Representative confocal images of Sac1-degron A431 cells treated with DMSO (4 h) or IAA for 2 or 4 h and stained for GM130 (*cis*-Golgi) or TGN46 (*trans*-Golgi). **e** Bar graph depicting the percentage (mean ±S.E.M) of cells with intact, partially fragmented, or fully fragmented Golgi. n = 3 independent experiments. **f** Western blot analysis of Sac1-degron A431 cells treated with DMSO or IAA for 4 h and blotted for TGN46, Golgin-97, GM130 and GRASP55. **g** Representative TEM images of Sac1-degron A431 cells treated with DMSO or IAA for 2 h. The yellow arrows indicate the Golgi stacks.

Fig. 4c). However, when we applied EN6, an activator of vacuole-type ATPase (V-ATPase)[35] to counteract the pH elevation, the TGN pH dropped to ~6.0 in Sac1-depleted cells (Fig. 3b). Although EN6 marginally affected inducible Sac1 depletion, re-acidification of the TGN efficiently prevented Sac1 depletion induced B4GALT1 and TGN46 degradation (Fig. 3c, d), and partially restored the TGN morphology

(Fig. 3e, f). These data argue that acute Sac1 depletion deacidifies the TGN, thereby inducing selective TGN morphological changes and degradation.

V-ATPase is a ubiquitous and essential regulator of organelle pH, with the cytoplasmic $V_1$ region consuming ATP to facilitate proton pumping through the membrane-embedded $V_0$ region. The reversible

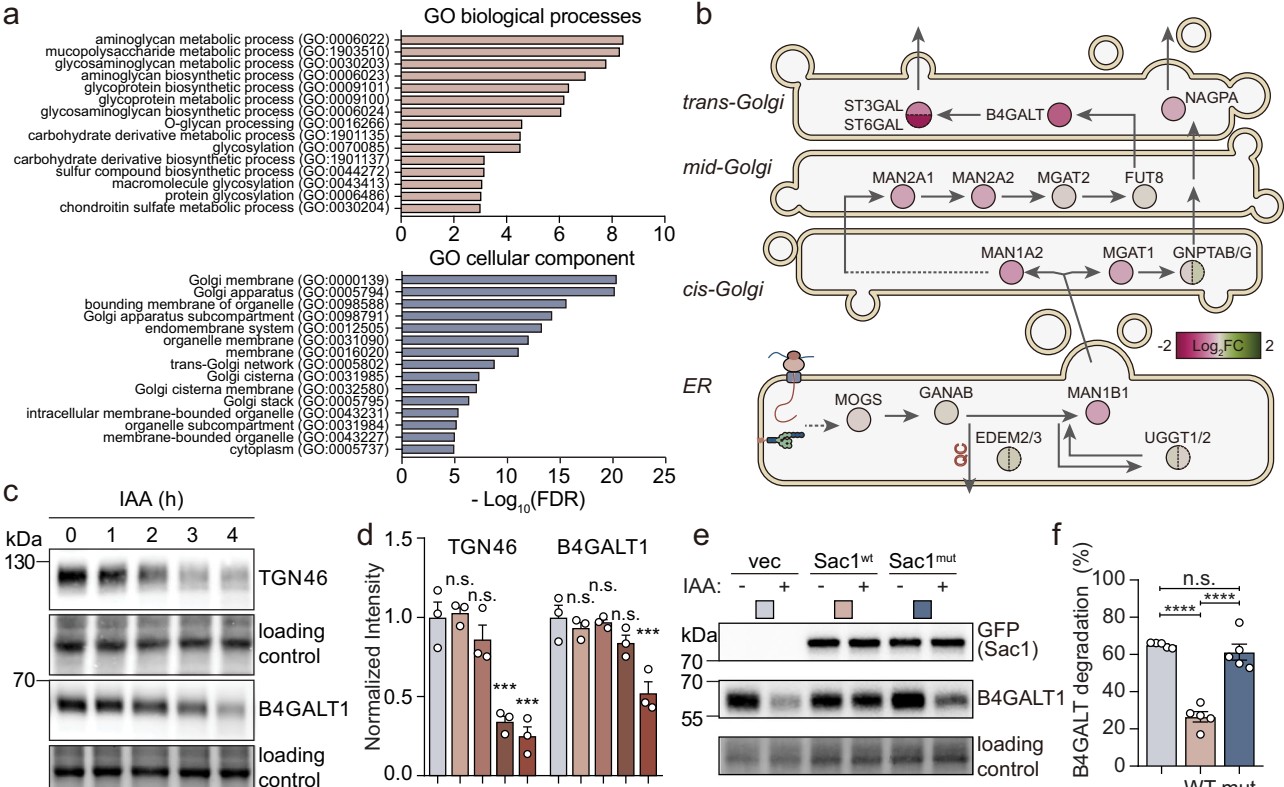

**Fig. 2 | Terminal glycosylation is impaired by Sac1 depletion. a** Top terms in gene ontology enrichment analysis of the downregulated proteins in Sac1-degron A431 cells treated with IAA for 3 h compared to DMSO control (-Log$_{10}$ False Discovery Rate (FDR) with a fold change threshold of ≥ 1.30 or ≤ 0.77 and an adjusted p-value of ≤ 0.05). **b** Schematic overview illustrating the changes in protein levels of glycosylation enzymes in Sac1-degron A431 cells after 3 h IAA treatment. The Log$_2$ fold change (Log$_2$FC) of individual enzyme is presented using a color gradient from magenta (−2) to green (+2). Glycosylation starts in the ER, where precursor glycan chains are added to the protein. The glycans are modified sequentially by glyco-sidases and glycosyltransferases along the ER and Golgi stacks. **c** Western blot analysis of TGN46 and B4GALT1 in Sac1-degron A431 cells treated with IAA for 0 to

4 h. **d** Bar graph showing normalized intensity of B4GALT1 and TGN46 (mean ±S.E.M) quantified from (**c**). n = 3 independent experiments. One-way ANOVA with Dunnett's multiple comparisons test. TGN46: n.s. p = 0.9964, p = 0.5307, *** p = 0.0004, p = 0.0001. B4GALT1: n.s. p = 0.8336, p = 0.9872, p = 0.2245, ** p = 0.0006. **e** Western blot analysis of lysates of HEK293A Sac1-degron cells transfected with pcDNA3.1 (empty vector), EGFP-Sac1-WT (Sac1$^{wt}$) or EGFP-Sac1-C389S (Sac1$^{mut}$), treated for 6 h with DMSO or IAA and blotted for B4GALT1 and GFP (Sac1). **f** Graph depicts the B4GALT1 degradation efficiency (mean ±S.E.M) in Sac1-degron HEK293A cells treated for 6 h with IAA and rescued with Sac1$^{wt}$ or Sac1$^{mut}$. n = 5 independent experiments. One-way ANOVA with Tukey's multiple compar-isons test. n.s. p = 0.6379, ****p < 0.0001.

---

assembly and disassembly of the V$_0$ and V$_1$ regions orchestrates proton translocation to adapt to various physiological demands (Fig. 3g)[36]. We therefore studied if the V-ATPase complex is affected by Sac1 depletion. The total amounts of V$_0$- and V$_1$-subunits were not altered by acute Sac1 degradation, as assessed by specific antibodies (Supplementary Fig. 4d). However, membrane-cytosol fractionation sug-gested that upon 2 h of Sac1 degradation the membrane-association of the V$_1$A subunit was reduced (Supplementary Fig. 4e, f).

We next studied the subcellular localization of the V$_1$ region. For this, we employed Sac1-degron A549 cells, due to the prominent cytoplasmic V$_1$A immunoreactivity in A431 cells that complicated the assessment. We found substantial colocalization of the V$_1$A subunit with the Golgi-specific V$_0$a2 subunit in control conditions, as expected[37]. Remarkably, this colocalization diminished gradually dur-ing 2 – 4 h after inducing Sac1 degradation (Fig. 3h, i). Co-staining with Golgin-97 (Fig. 3j, k) and LAMP2 (Fig. 3l, m) antibodies revealed that at 2 h after inducing Sac1 degradation, the TGN localization of the V$_1$ region was reduced, while its lysosomal localization was maintained. In parallel, the Golgi specific V$_0$a2 subunit clearly overlapped with the Golgi marker GM130, while no colocalization with the lysosomal marker LAMP2 was observed (Supplementary Fig. 4g). Together, these data suggest that the V$_0$ and V$_1$ regions of the Golgi V-ATPase complex become disassembled rapidly after Sac1 depletion, providing a mechanism for the deacidification of the TGN.

## Sac1 controls V-ATPase complex integrity via its lipid environment

As Sac1 fuels PI(4)P/cholesterol lipid exchange by lipid transfer pro-teins, i.e. OSBP[16,17,20], we assessed the role of Sac1 in controlling PI(4)P and cholesterol distribution in the TGN. Consistently with the mass spectrometry results (Fig. 1c), expression of mCherry-P4M-SidM and immunostaining with PI(4)P antibody in A431 Sac1-degron cells revealed an increasing PI(4)P signal over time, starting perinuclearly at early time points after degradation and extending later on throughout the cell (Supplementary Fig. 5a–c). The latter agrees with findings from chronic Sac1 depletion systems[10,21,24]. Remarkably, already at 1 h of Sac1 degradation, we observed a prominent accumulation of PI(4)P in the TGN region (Fig. 4a, b). Staining of live cells with mNeonGreen-ALOD4, recognizing PM accessible cholesterol[38], showed that after 1 h of Sac1 removal the PM accessible cholesterol levels were slightly reduced, and by 4 h, only ~20% of control levels remained (Supplementary Fig. 5d, e). Filipin staining and thin-layer chromatography showed a minor reduction in total cellular free cholesterol over time (Supple-mentary Fig. 5f–h). However, by 2 h of Sac1 degradation, the filipin signal in the TGN area was significantly reduced (Fig. 4c, d) and in the lysosomes slightly increased (Supplementary Fig. 5i, j). Meanwhile, cholesterol esterification was increased 2.5-fold by 2 h (Supplementary Fig. 5k) and total cholesteryl ester content elevated 1.5-fold by 4 h (Supplementary Fig. 5l), likely due to reduced cholesterol delivery

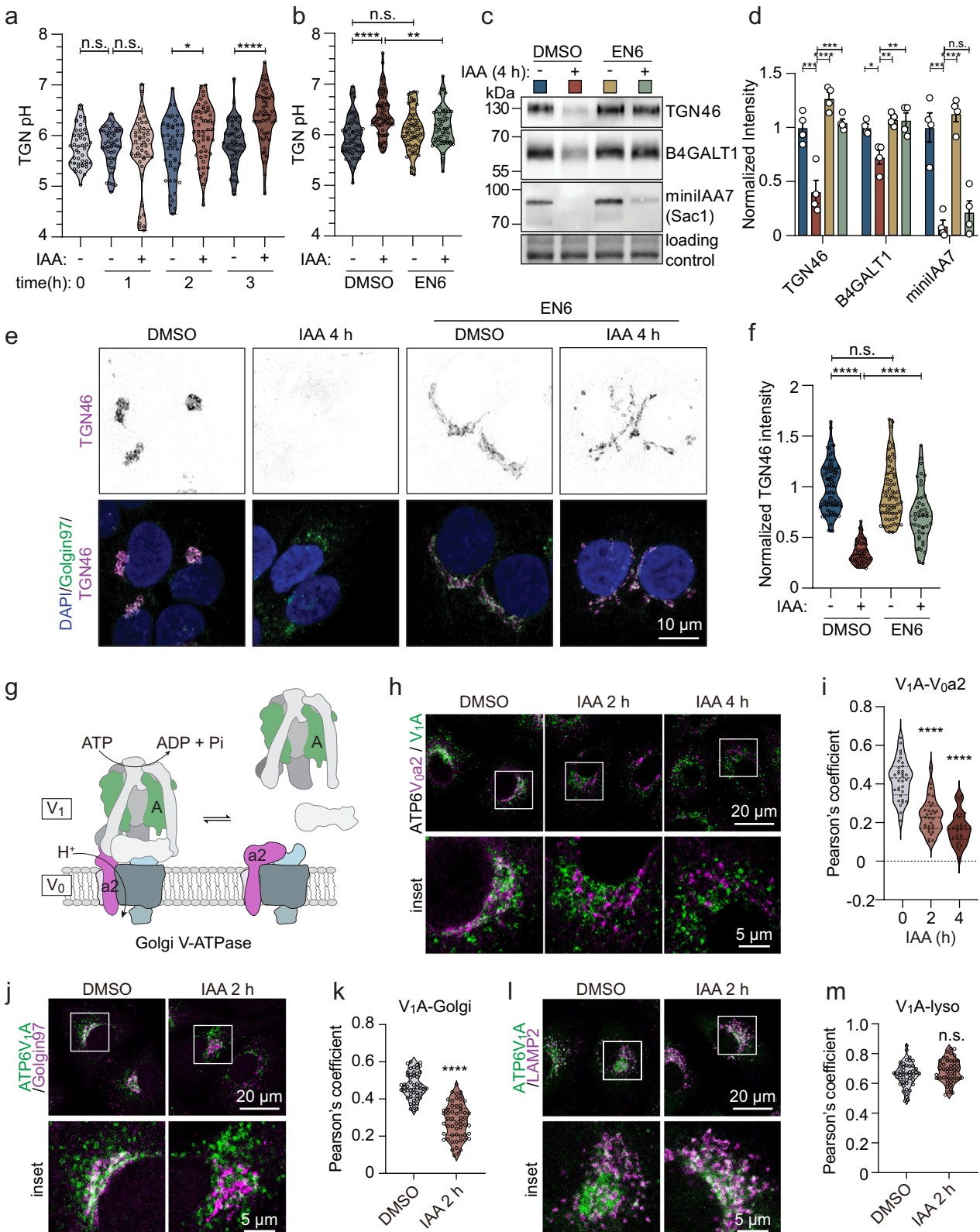

from the ER to the TGN, resulting in cholesterol accumulation in the ER. Together, these results provide evidence that degradation of Sac1 rapidly alters both the PI(4)P and cholesterol content of the TGN.

Next, we investigated if the changes in the TGN lipid environment induced by Sac1 degradation affect the Golgi V-ATPase by using sucrose density gradient fractionation of cellular post-nuclear supernatants (PNS). We found that the floating Golgi V-ATPase $V_0$

region peaked in the light membrane fraction 2, as probed by an anti-$V_0a2$ antibody (Fig. 4e, f). Upon Sac1 depletion for 2 h, the flotation of the $V_0$ region became less efficient, suggesting that the membrane environment of the $V_0$ region shifted towards higher equilibrium density (Fig. 4e, f). Interestingly, cellular cholesterol depletion phenocopied the effect of Sac1 depletion on $V_0$ flotation (Supplementary Fig. 6a), suggesting that the cholesterol reduction in the TGN

**Fig. 3 | Acute Sac1 depletion results in TGN deacidification caused by V-ATPase disassembly. a** Violin plot showing the distribution of TGN pH values of Sac1-degron A431 cells transiently transfected with GalT-RpHLuorin2 and treated with DMSO or IAA for 0 to 3 h. n = 3 independent experiments. One-way ANOVA with Tukey's multiple comparisons test. n.s. p > 0.9999, p = 0.9848, * p = 0.0480, **** p < 0.0001. **b** Violin plot depicting the distribution of TGN pH values of Sac1-degron A431 cells treated for 3 h with DMSO or IAA with or without 50 μM EN6. n = 3 independent experiments. One-way ANOVA with Tukey's multiple comparison test. n.s. p = 0.4775, ** p = 0.0039, **** p < 0.0001. **c** Western blot analysis of Sac1-degron A431 cells treated for 4 h with DMSO or IAA with or without 50 μM EN6. Immunoblotted for TGN46, B4GALT1 and miniIAA7 (Sac1). **d** Bar graph showing average normalized immunoblot intensities from (**c**). n = 4 independent experiments. One-way ANOVA with Tukey's multiple comparisons test. TGN46: *** p = 0.0004, p = 0.0002, **** p < 0.0001. B4GALT1: * p = 0.0127, ** p = 0.0022, p = 0.0026. mini-IAA7: *** p = 0.0001, **** p < 0.0001. n.s. p = 0.7763. **e** Representative confocal images of Sac1-degron A431 cells treated for 4 h with DMSO or IAA with or without 50 μM EN6. Stained for DAPI, Golgin-97 and TGN46. **f** Violin plot depicts the distribution of the TGN46 intensity in the Golgi area calculated from (**e**). n = 3 independent experiments. One-way ANOVA with Tukey's multiple comparisons test. n.s. p = 0.6182, **** p < 0.0001. **g** Cartoon illustrating the reversible assembly of the $V_0$ and $V_1$ regions of the V-ATPase complex. Upon assembly, ATP hydrolysis of the $V_1$ region drives the proton pumping ability of the $V_0$ region. **h** Representative fluorescence micrographs of Sac1-degron A431 cells treated with DMSO for 4 h or IAA for 2 or 4 h and stained for ATP6$V_0$a2 or ATP6$V_1$A. **i** Violin plot showing the distribution of the Pearson's correlation coefficient between ATP6$V_0$a2 and ATP6$V_1$A from (**h**). Representative data from one of three different experiments. One-way ANOVA with Dunnett's multiple comparisons test. **** p < 0.0001. Representative confocal images of Sac1-degron A549 cells treated with DMSO or IAA for 2 h and stained for ATP6$V_1$A and Golgin-97 (**j**) or ATP6$V_1$A and LAMP2 (**l**). **k, m** Violin plot showing the Pearson's correlation coefficient between ATP6$V_1$A and Golgin-97 (**j**) and ATP6$V_1$A and LAMP2 (**l**). n = 3 independent experiments. Two-tailed unpaired Student's t-test. **** p < 0.0001, n.s. p = 0.1287.

membrane contributes to the behavior of the $V_0$ region in Sac1 depleted cells.

To probe this more directly, we replenished cells with cholesterol while Sac1 was being degraded. This effectively rescued the flotation of the $V_0$ region (Fig. 4e, f). We also tested the effect of inhibiting PI(4)P generation by PI4K inhibitors (PIK93 for PI4KIIIβ and NC03 for PI4KIIα), the main PI4Ks in the Golgi[6], during Sac1 degradation. This did not revert the $V_0$ region to low-density membrane fractions (Fig. 4e, f), suggesting that PI(4)P, as a minor membrane constituent, does not affect the overall flotation behavior of the $V_0$ region. Together, these findings suggest that cholesterol reduction in the TGN induced by Sac1 removal affects the TGN membrane environment of the $V_0$ region.

Next, we analyzed if cholesterol supplementation or PI4K inhibitor treatment during Sac1 degradation can rescue $V_0$-$V_1$ complex assembly. Interestingly, both increasing membrane cholesterol or decreasing PI(4)P generation improved the $V_0$-$V_1$ colocalization in Sac1 depleted cells (Fig. 4g, h), while only PI4K inhibition could partially rescue Golgi morphology (Supplementary Fig. 6b–d). Collectively, the data suggest that Sac1 degradation alters the TGN membrane lipid composition resulting in V-ATPase complex disassembly, providing a mechanism for the TGN deacidification and glycosylation defects.

## The conformation of the $V_0$ region is influenced by the membrane lipid composition

To investigate how the membrane lipid composition affects the Golgi V-ATPase complex, we carried out atomic-level molecular dynamics simulations of the $V_0$ region containing the Golgi specific $V_0$a2 subunit (Fig. 5a), focusing on the role of PI(4)P and cholesterol (Fig. 5b and Table 1). We analyzed the simulation data using deep learning techniques, looking for structural classes where the $V_0$ region conformations are similar. Using this approach, the conformational space was first reduced to two dimensions using an autoencoder, after which these states were clustered with a Bayesian Gaussian Mixture Model, searching for structural classes representing distinct conformations. The analysis revealed four separate clusters (Fig. 5c) characterized by balanced populations (Supplementary Fig. 7a) and high structural similarity within each cluster (Supplementary Fig. 7b). A representative conformation of the $V_0$ region from each of the four clusters is depicted in Fig. 5d (top view) and Supplementary Fig. 7c (side view). Based on structural data of the $V_0$-$V_1$ complex, the assembly of the $V_0$ and $V_1$ regions requires the detachment of the cytosolic head domain of the $V_0$a2 subunit from the d1 subunit, otherwise the a2 head sterically hinders $V_0$-$V_1$ complex assembly. Similar structural rearrangements of the yeast $V_0$a cytosolic domain have been described to be essential for $V_0$-$V_1$ assembly/disassembly[39].

Analysis of the simulation data showed that among the four conformational clusters detected, cluster 2 plays a crucial role in the assembly process, as in these conformations the a2 head formed the fewest interactions with the d1 subunit (Fig. 5e). Thus, cluster 2 characterizes conformations of the $V_0$ region that are prone to bind to the $V_1$ region to form a functional $V_0$-$V_1$ complex. Notably, these conformations differ from those in the other clusters (Fig. 5f). PLS-based discriminant analysis of the simulation data (Supplementary Movie 1, 2) showed that, compared to the other clusters, in cluster 2 the arm and head of the a2 subunit are positioned close to the membrane and do not sterically interfere with the binding orientation of the $V_1$ region. The $V_0$a2 head can interact with PI(4)P in the membrane, as described previously[37]. In line with this, our data show that in cluster 2 the number of interactions between the a2 head and PI(4)P molecules in the membrane is the highest (Fig. 5g).

## High level of PI(4)P suppresses the conformational state of the $V_0$ region that promotes $V_0$–$V_1$ complex assembly

We next analyzed the probability of occurrence of cluster 2 in membranes of different lipid compositions (Fig. 5h). We found that the cluster 2 population is minimized (<10% of conformations) when a cholesterol-free membrane is enriched with ample (20 mol%) PI(4)P. When 10 mol% cholesterol is added to this membrane, the population of cluster 2 increases to ~24%. The population of cluster 2 is at its highest (40%) when the concentration of PI(4)P is reduced to 10 mol% and in parallel cholesterol is increased to 20 mol%. At 10 mol% PI(4)P and 20 mol% cholesterol, the $V_0$a2 head has the least interactions with the d1 subunit (Fig. 5i), thus supporting a conformation that can form a complex with the $V_1$ region. These data imply that in the absence of cholesterol, high PI(4)P suppresses the assembly-promoting conformational state of $V_0$ region, while cholesterol plays an assembly-promoting, rescuing role.

## Cholesterol blocks the access of PI(4)P deep into the pocket between the a2-subunit and the c-ring

From the atomic-level simulation data, we elucidated the mechanisms of PI(4)P and cholesterol action by focusing on their densities in the immediate vicinity of the $V_0$ region (Supplementary Fig. 7d). In membranes with high PI(4)P and without cholesterol (mimicking Sac1 degradation), we found that PI(4)P in the cytosolic membrane leaflet occupies the pocket between the a2 body and the c-ring, where it can interact with the a2 arm, a2 body, and the c-ring (Fig. 5j and Supplementary Fig. 7d). In cholesterol and PI(4)P containing membranes, cholesterol prevents PI(4)P from entering the pocket between the a2 body and the c-ring (Fig. 5j and Supplementary Fig. 7d), promoting an active conformation of the $V_0$ region. Under these conditions, potential salt bridges can be observed in the pocket between the a2 subunit and the c-ring [between E81 (a2 arm) – R436 (a2 body) and D76 (a2 arm) – R126 (c-ring), Supplementary Fig. 6e, f]. While these salt bridges appear most frequently in cluster 2 conformations (Supplementary

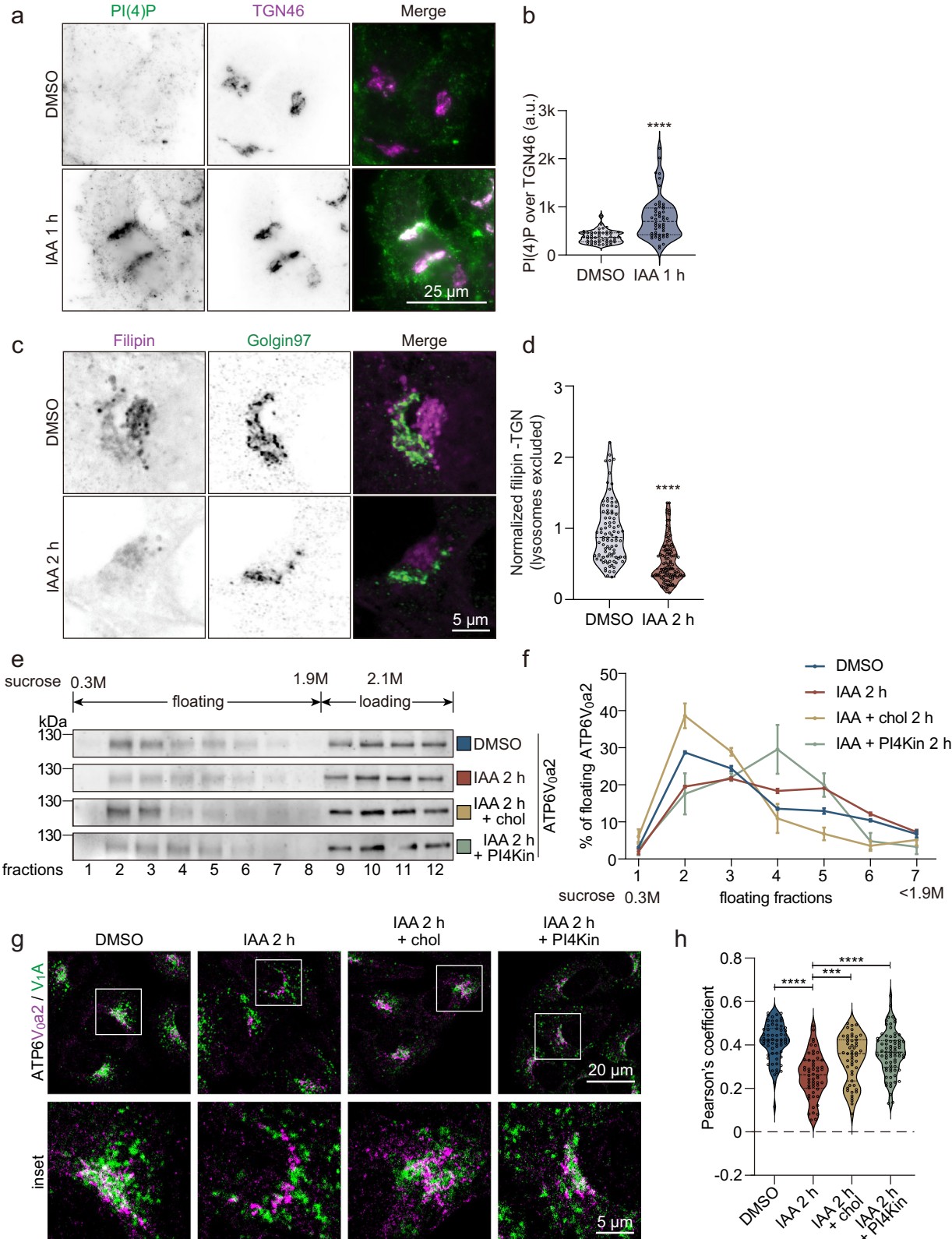

Fig. 7e, f), their role(s) in the conformational changes of the $V_0a2$-head remain to be elucidated.

To further investigate how the lipid dependent conformational changes of the $V_0a2$ arm affect $V_0$-$V_1$ complex formation, we assessed the compatibility of the $V_0$ sector to form a docked $V_0$-$V_1$ complex. We fitted the simulated $V_0$ conformations to the cryo-EM structure[40] and counted the number of clashes, i.e. atom-atom contacts, between the

simulated $V_0$ structure and the $V_1$ region of the cryo-EM structure (Fig. 5k). This showed that the active conformation promoting cluster 2 has the lowest number of clashes between the $V_0$ and $V_1$ sectors (Fig. 5l), as would be expected. Moreover, a combination of 10 mol% PI(4)P with 20 mol% cholesterol in the membrane is associated with the least clashes (Fig. 5m), thus favoring $V_0$-$V_1$ complex assembly. Together, these data suggest that the Golgi V-ATPase assembly is highly

**Fig. 4 | Sac1 fuelled PI(4)P/cholesterol exchange maintains V-ATPase activity.**
**a** Representative confocal images of Sac1-degron A431 cells treated with IAA or DMSO for 1 h and stained for PI(4)P and TGN46. **b** Violin plot showing the average PI(4)P intensity in the Golgi area (TGN46). Representative data from one of three independent experiments. Two-tailed unpaired Student's t-test. **** $p < 0.0001$. **c** Representative confocal images of Sac1-degron A431 cell treated with IAA or DMSO for 2 h and stained with Filipin and Golgin-97. **d** Violin plot showing the normalized filipin intensity in the Golgi area (Golgin-97). Data from 3 independent experiments. Two-tailed Mann-Whitney U test. **** $p < 0.0001$. **e** Western blot analysis of sucrose gradient fractionations (Sucrose concentration from 0.3 M (fraction 1) to 2.1 M (loading fractions 9 – 12)) of A549 Sac1-degron cells treated for 2 h with DMSO, IAA, IAA with cholesterol, or IAA with PI4K inhibitors (NC03 and PIK93) and blotted for ATP6V$_0$a2. **f** Line graph depicting the average percentage (mean±S.E.M.) of floating ATP6V$_0$a2 distributed between fraction 1 to 7 of A549 Sac1-degron cells treated for 2 h with DMSO, IAA, IAA plus cholesterol, or IAA plus PI4K inhibitors (NC03 and PIK93). n = 3 technical replicates. **g** Representative confocal images of A549 Sac1-degron cells treated for 2 h with DMSO, IAA, IAA in combination with cholesterol or PI4K inhibitors. Stained for ATP6V$_1$A and ATP6V$_0$a2. **h** Violin plot depicting the distribution of the Pearson's correlation coefficient between ATP6V$_1$A and ATP6V$_0$a2 from (**g**). Data from one of three independent experiments. One-way ANOVA with Tukey's multiple comparisons tests. *** $p = 0.0008$, **** $p < 0.0001$.

sensitive to its membrane environment. In membranes mimicking the Golgi lipid environment upon Sac1 degradation, PI(4)P accumulates in the pocket between the a2 body and the c-ring, due to insufficient cholesterol to block it. This reduces the propensity to reach a V$_0$ conformation that is capable of associating with the V$_1$ region.

### Sac1 safeguards hCG processing in human trophoblasts

Our experimental findings identified an essential role for Sac1 in maintaining Golgi morphology and terminal glycosylation via V-ATPase mediated pH regulation. We next challenged the physiological relevance of this idea. As Sac1 loss in mouse is embryonically lethal[22,28], variations in the *SACM1L* gene might not be well tolerated in humans. Indeed, based on the Genome Aggregation Database (gnomAD) the probability of the *SACM1L* gene for being loss-of-function intolerant (lof.pLI) is high, 0.97 (Fig. 6a). Additionally, the observed number of loss-of-function *SACM1L* variants is only 27% of the expected (o/e = 0.27, 90% CI: 0.15-0.51), consistent with negative selection acting on these variants in humans and with an essential role of Sac1 also in human development.

During the early stages of mammalian embryogenesis, secretion of growth factors, hormones and cytokines orchestrates close communication between the blastocyst and the uterine wall. Particularly, human chorionic gonadotropin (hCG), a trophoblast-secreted glycoprotein, is required for implantation and gestation[41]. To investigate if Sac1 is important for hCG secretion by trophoblasts, we differentiated human embryonic stem cells (hESCs) carrying the Sac1-degron system[30] into trophoblasts (TB) (Fig. 6b). After 10 days of differentiation, the trophoblast markers hCGβ (beta-subunit of hCG), GATA3 and Dab2 were readily detected (Supplementary Fig. 8a), and multinucleated syncytiotrophoblasts (STB) were obtained after an additional 6 days of differentiation. Importantly, the levels of hCGβ increased drastically during differentiation, as anticipated (Supplementary Fig. 8b).

Acute depletion of Sac1 from trophoblasts led to a severe fragmentation of the Golgi by 4 h (Fig. 6c), followed by reduced protein levels of TGN46 and B4GALT1 (Supplementary Fig. 8c, d). Remarkably, when EN6 was added to the trophoblasts together with IAA, the TGN46 protein levels could be restored and B4GALT1 levels were partially rescued (Fig. 6d, e). At 1-2 h of Sac1 degradation, PI(4)P levels rapidly increased in the TGN (Fig. 6f and Supplementary Fig. 8e), while TGN cholesterol levels decreased (Fig. 6g and Supplementary Fig. 8f, g). Sac1 degradation in trophoblasts also acutely induced V-ATPase disassembly within 2 h (Fig. 6h).

Next, we assessed the impact of Sac1 depletion on trophoblast hCGβ processing and secretion. At 15 h upon Sac1 removal, the cellular levels of hCGβ were increased and the highly glycosylated hCGβ, abundantly secreted in normal conditions, was massively reduced in the medium (Fig. 6i, j). Neuraminidase treatment of hCGβ increased the antibody reactivity and showed that hCGβ was abundantly sialylated in control trophoblasts and that some residual sialylation took place in Sac1 depleted cells. However, upon Sac1 degradation, hCGβ with a lower molecular weight was secreted, implicating a glycosylation defect (Fig. 6j). Overall, this demonstrates that key phenotypes observed upon acute Sac1 degradation are recapitulated in differentiated trophoblasts and highlights the Golgi V-ATPase complex as a key effector of Sac1 activity that safeguards hCGβ processing and secretion in trophoblasts.

## Discussion

This study employs the potential of AID in elucidating the early functional consequences of eliminating a vital protein, Sac1. The rapid, tightly controlled degradation of Sac1 enabled us to systematically follow the time course of ensuing defects, thereby revealing underlying causalities. In approximately one hour, Sac1 was fully degraded, followed by a ~3-fold increase in PI(4)P concentrated in the TGN area and a decrease in free cholesterol levels in the TGN during the following hour, likely due to OSBP inhibition. These changes in the TGN lipid environment prevent the assembly of the Golgi V-ATPase complex as observed from 2 h onwards, subsequently elevating TGN pH from ~5.9 to ~6.3 and resulting in disruption of terminal glycosylation and selective TGN degradation by 4 h (Fig. 7). Mechanistically, the changes in the lipid environment of the TGN upon acute Sac1 depletion induce conformational changes in the V-ATPase V$_0$ region, inhibiting V$_0$-V$_1$ complex assembly. Importantly, the key phenotypes ensuing from Sac1 degradation could be recapitulated in human differentiated trophoblasts, resulting in defects in hCGβ glycosylation and secretion. This may in part explain the preimplantation lethality of Sac1 null mice and rationalizes the intolerance for Sac1 loss-of-function during mammalian development.

The most direct effect of loss of Sac1 function is the accumulation of its ligand PI(4)P that facilitates cargo trafficking from the TGN[10,11,21,42]. Importantly, we provide evidence that Sac1 is needed not only to maintain PI(4)P levels, but also the high cholesterol content in the TGN, in line with the critical role of Sac1 in fuelling PI(4)P/cholesterol exchange by OSBP at ER-TGN contact sites[11,16,17,43]. As complex interactions between lipid transfer proteins control cholesterol and PI(4)P levels in the TGN[20], Sac1 most likely affects a variety of lipid transfer proteins beyond OSBP. The upregulation of several lipid transporters, observed in our proteomics data, probably serves to compensate for the acute loss of Sac1. However, this is clearly insufficient to maintain the normal ER-TGN (and ER-plasma membrane) cholesterol gradient without Sac1 burning off PI(4)P. Hence, the critical role of Sac1 in governing cellular cholesterol distribution contributes to its vital importance. Interestingly, Sac1 can interact with the ER cholesterol sensor SCAP, suggesting an intricate coregulation between Sac1 fuelled PI(4)P/cholesterol exchange and transcriptional control of cholesterol homeostasis[10]. Besides cholesterol, PI(4)P is involved in counter exchange with other lipids, such as phosphatidylserine[44]. Thus, over time Sac1 depletion will challenge lipid homeostasis broadly.

The present study highlights the TGN membrane lipid dependent control of Golgi V-ATPase activity via Sac1 as an early, shared denominator and driver for the later phenotypes, i.e. Golgi deacidification, fragmentation and degradation, as well as cargo glycosylation and secretion defects. This notion is supported by the observations that TGN fragmentation, protein degradation and defective cargo secretion could be partly rescued by pharmacological re-acidification.

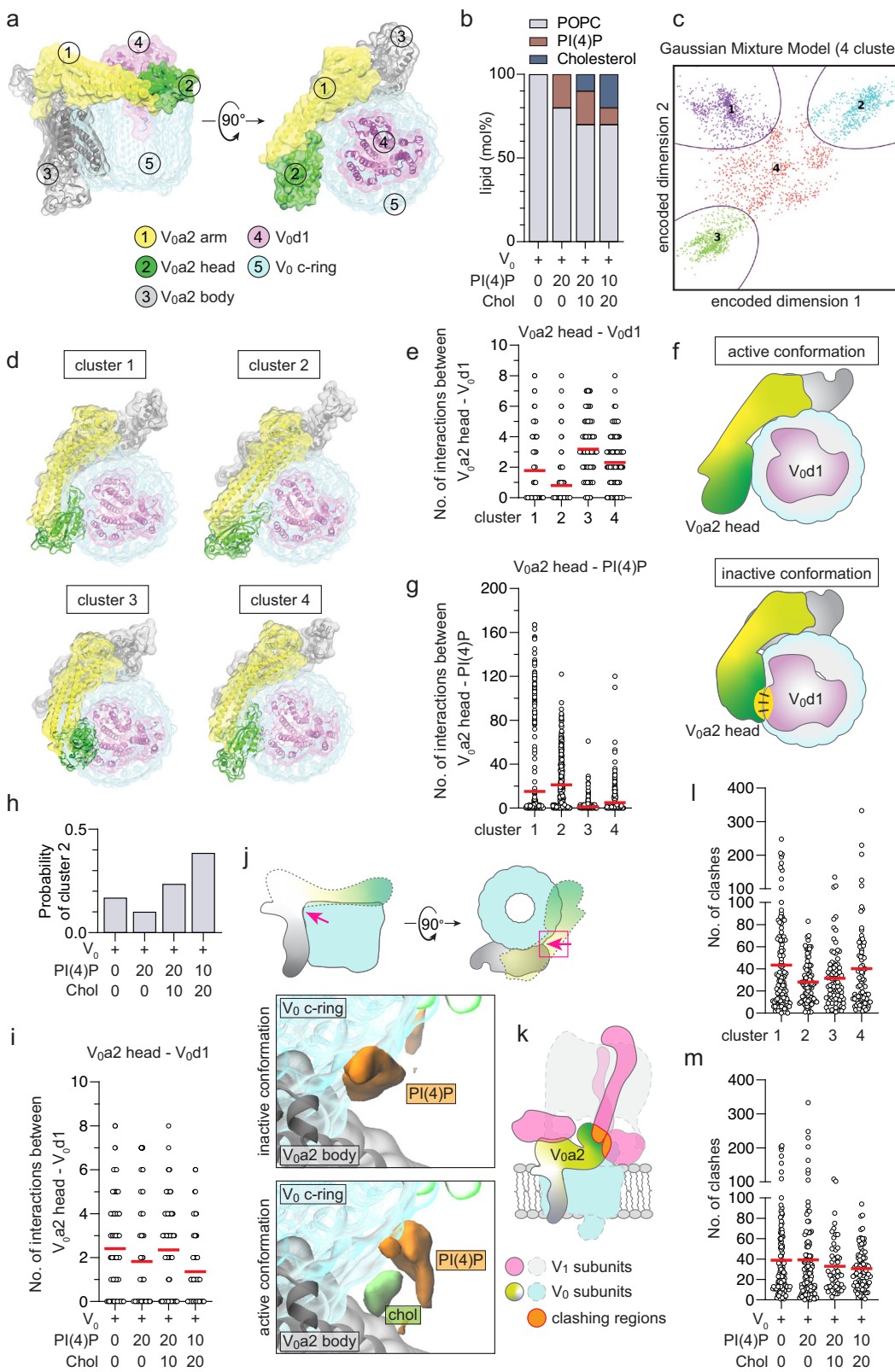

**Nature Communications** | (2025)16:7808

It also agrees with the established role of V-ATPase in safeguarding Golgi glycosylation and secretion[45–50]. Interestingly, mutations in $V_0a2$ are also linked to trafficking and glycosylation defects and can lead to connective tissue disorders, such as wrinkly skin syndrome and cutis laxa[45,51,52]. As Sac1 activity has also been implicated in cargo secretion from the TGN[23,24,53], the coupling of TGN carrier formation to V-ATPase activity may help to coordinate terminal glycosylation and secretion.

V-ATPases are key regulators of organelle acidification conserved from plants to humans. The regulation of V-ATPases has been best characterized in *Saccharomyces cerevisiae*, where the assembly/disassembly of $V_0$ and $V_1$ regions rapidly adjusts proton pumping activity, with e.g. glucose starvation stimulating complex disassembly and conservation of energy[54]. In mammalian cells, the cues controlling the reversible association of $V_0$ and $V_1$ regions may be more context

**Fig. 5 | Atomic-level molecular dynamics simulations of the $V_O$ region in different lipid membrane compositions. a** Top and side view of the Golgi V-ATPase $V_O$ region. The $V_O$a2 subunit is divided into three regions: a2-body (grey), a2-arm (yellow), and a2-head (green). The d1 subunit is shown in magenta. The c-ring is shown in cyan. **b** Graphical representation of the different membrane lipid compositions used in the simulations. **c** Clustering of the simulation data using Bayesian Gaussian Mixture Models (BGMM) reveals four distinct clusters, each representing a different structural class. **d** Representative top view of the $V_O$ region from each cluster. For each cluster, the structure with the smallest distance from the mean of the Gaussian in the BGMM is presented. **e** Scatter plot (red line = mean) showing number of interactions formed by the $V_O$a2-head with the $V_O$d1 subunit in each cluster. **f** Schematic representation of the active conformation (cluster 2) and the inactive conformations, where the $V_O$a2-head interacts with the $V_O$d1 subunit. **g** Scatter plot (red line = mean) showing number of interactions between the $V_O$a2- head and PI(4)P molecules in each cluster. **h** Probability (relative population) of cluster 2 with each membrane lipid composition. **i** Scatter plot (red line = mean) showing number of interactions between the $V_O$a2-head and the $V_O$d1 subunit in each lipid composition. **j** Occupational density maps of PI(4)P (orange) and cholesterol (green) densities in the vicinity of the $V_O$ region and c-ring in a representative inactive and active conformation from a top view. Insets from representative images shown in Supplementary Fig. 6d. Red arrows indicate the pocket between the a2 subunit and the c-ring. **k** Schematic representation of the potentially clashing regions between the $V_O$a2 arm and the $V_1$ region. The $V_O$ region is based on the atomistic simulations data, which is fitted one by one on the cryo-EM structure 6WM2 of the human V-ATPase. **l** Scatter plot (red line = mean) showing number of clashes between the $V_O$a2-arm from each cluster with the $V_1$ region. **m** Scatter plot (red line = mean) showing number of clashes between the $V_O$a2-arm from each lipid composition with the $V_1$ region.

**Table 1 | Description of the simulated model systems**

| Simulation model | POPC (mol%) | PI(4)P (mol%) | Cholesterol (mol%) | Simulation time (ns) | Replicas |
|---|---|---|---|---|---|
| $V_O$ | 100 | 0 | 0 | 750 | 8 |
| $V_O$–20PI4 | 80 | 20 | 0 | 750 | 8 |
| $V_O$–20PI4-10Chol | 70 | 20 | 10 | 750 | 8 |
| $V_O$–10PI4-20Chol | 70 | 10 | 20 | 750 | 8 |

dependent, with e.g. glucose removal promoting either assembly or dissociation[36,55], and studies have mainly focused on the lysosomal V-ATPase[56–59].

Despite V-ATPases being large integral membrane protein complexes, there is limited understanding on how they are functionally regulated by lipids. In yeast, V-ATPase activity can be modulated by phosphoinositides, phospholipids and ergosterol[37,60–63]. The human $V_O$a-subunits were shown to bind to distinct phosphoinositides, with the endo-lysosomal isoform binding to PI(3)P and PI(3,5)$P_2$ and the Golgi isoform to PI(4)P in vitro[37,64,65]. Importantly, the cryo-EM structures of human V-ATPase reveal several lipids, including phospholipids and cholesterol, as integral parts of the $V_O$ region[40,66].

The present study complements and extends these findings by identifying Sac1 as a gatekeeper whose activity controls, via effects on the TGN membrane lipid environment, the conformation and activity of Golgi V-ATPase in the cellular context. Specifically, we provide evidence that the dissociation of the $V_O$ and $V_1$ regions is promoted by increased PI(4)P and decreased cholesterol in the TGN membrane upon Sac1 removal. Atomic-level simulations illuminate the mechanistic role of cholesterol in stabilizing the assembly-promoting conformation of the $V_O$ region. In the absence of cholesterol, PI(4)P can move into the recess between the a2 subunit and the c-ring of the $V_O$ region and thus hinder the assembly-promoting conformation. The assembly-disrupting conformations lead to binding of the a2 head to the d1 subunit, which sterically prevents the $V_1$ region from assembling with the $V_O$ region. Cholesterol, on the other hand, prevents PI(4)P from entering deep into this recess, thereby promoting assembly.

Based on our data, the V-ATPase assembly status was more sensitive to loss of Sac1 function at Golgi than lysosomal membranes: upon 2 h of Sac1 removal - when PI(4)P was the only phosphoinositide increased – the $V_1$ region became disassembled from the Golgi $V_O$ region but the localization of the $V_1$ region in lysosomal membranes appeared intact. Moreover, at 4 h of Sac1 removal, TGN46 degradation was still sensitive to lysosomal $V_O$-$V_1$ dissociation by BafA1. The early affliction of Golgi V-ATPase may be related to different phosphoinositide sensitivities of the V-ATPases, but possibly also to a predominant activity of Sac1 at Golgi contacts.

The initial effects of Sac1 removal that we focused on, centred on the TGN. Yet, other compartments, such as endo-lysosomes, are likely affected over time. Interestingly, recent data implicate lysosomal PI(4)

P, generated by local synthesis and turned over by Sac1, in lysosomal membrane repair[67,68] and rewiring of lysosomal degradation during starvation[69]. We found all the major PIPs to be elevated by 4 h of Sac1 removal. As PIPs are rapidly hydrolyzed and phosphorylated by specific kinases and phosphatases, the initial increase of PI(4)P is expected to result in the interconversion to other PIP species over time[70]. The subcellular localization of individual PIP species is important for organelle specific membrane properties[7], and its dysregulation leads to defects in migration, autophagy, endo-lysosomal trafficking and signaling via PIP binding proteins[69,71]. Thus, a broad PIP imbalance affecting multiple endomembranes and PIP dependent lipid transfer processes may contribute to the long-term effects of Sac1 depletion. Moreover, considering the coupling between PIP and cholesterol distribution, it is likely that cholesterol and PIPs co-operate in the regulation of several membrane proteins, including V-ATPase, in other compartments.

In conclusion, this study reveals that in human cells, the Golgi V-ATPase assembly status is dynamically modulated by membrane lipids, specifically by the TGN PI(4)P and cholesterol pools that are physiologically co-regulated by Sac1 activity. The multiple modes of lipid – V-ATPase interactions and how they regulate V-ATPase activity will be important topics for future work.

## Methods
### Cultivation of A431, HEK293A and A549 cells and generation of cell lines
A431 (ATCC CRL-1555) and HEK293A (Invitrogen R70507) cells were cultured in DMEM (high-glucose, Gibco, 21969) and A549 (ATCC CCL-185) cells were cultured in F-12 Nutrient Mixture (Gibco, 21127-022). Media was supplemented with 10% FBS, penicillin/streptomycin (100 U ml$^{-1}$) and L-glutamine (2 mM). Cells were cultured at 37 °C in 5% $CO_2$. Human Embryonic Stem cells (hESCs) (H9, WiCell, WIC-WA09-RB-001) were cultured in mTeSR Plus culture medium (Stem Cell Technologies, 100-0276) and plated on Matrigel (Corning, 356231) coated dishes, diluted 1:200 in DMEM/F-12 (Gibco, 31331-028). For passaging, hESCs were washed 2x with 1x PBS (Corning, 21-040-cv) and treated with 0.5 μM EDTA (Invitrogen, 15575020) for 5 min at room temperature (RT). Sac1-degron cell lines (A431, A549, HEK293A and hESCs) were generated as previously described[30].

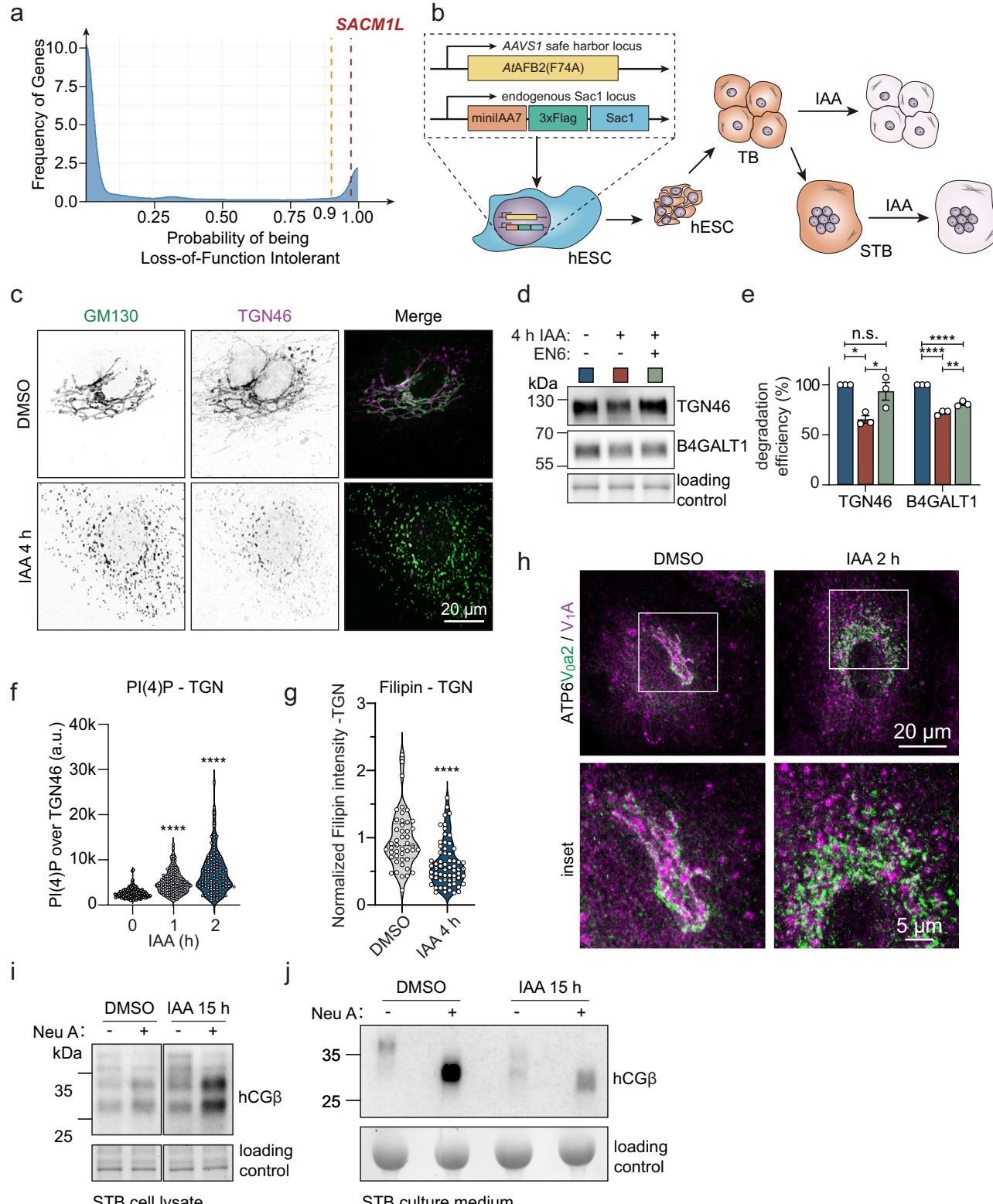

## hESC culture and trophoblasts differentiation

Wild-type and Sac1-degron H9 hESCs were maintained in mTeSR Plus medium on Matrigel-coated plates. Differentiation into trophoblasts was performed by using the differentiation protocol previously described[72]. In brief, hESCs were seeded at 1:4 ratio onto Matrigel-coated 6-well plates in the presence of 10 μM ROCK inhibitor (Selleckchem, Y-27632). The following day the cells were rinsed with DMEM/F-12 and transferred into TB differentiation medium (DMEM/F-

12 supplemented with penicillin/streptomycin 100 U ml$^{-1}$ each, 0.2% FBS, 0.3% fatty acid free BSA (Sigma-Aldrich, A-3803), 1% Insulin-Transferrin-Selenium-Ethanolamine (ITS-X) (Gibco, 51500056), 0.1 mM β-Mercaptoethanol (Roth, 4227.1), 1.5 μg ml$^{-1}$ L-ascorbic acid (Sigma-Aldrich, A92902), 50 ng ml$^{-1}$ EGF (PreproTech, AF-100-15), 2 μM CHIR99021 (Tocris, 4423), 0.5 μM A83-01 (Selleckchem, S7692), 1 μM SB431524 (Selleckchem, S1067), 0.8 mM Valproic acid (MedChemExpress, HY-10585), 5 μM ROCK inhibitor, and 10 ng ml$^{-1}$ BMP4 (R&D

**Fig. 6 | Sac1 depletion impairs hCG secretion in trophoblasts. a** Density plot of gnomAD dataset showing the frequency of genes (y-axis) of having the probability of being loss-of-function intolerant (lof.pLI, x-axis). The orange dotted line indicates the 0.9 threshold commonly used for Mendelian diseases and the red dotted line the location of the *SACM1L* gene. **b** Schematic overview of differentiation of trophoblasts from Sac1-degron hESCs. Sac1-degron hESCs are subsequently differentiated into trophoblasts (TB) and syncytiotrophoblasts (STB). IAA treatment can induce Sac1 degradation in both TB and STB. **c** Representative confocal images of Sac1-degron STB treated for 4 h with IAA and stained for GM130 and TGN46. **d** Western blot analysis of Sac1-degron STB treated for 4 h with DMSO or IAA and 50 μM EN6 and blotted for TGN46 and B4GALT1. **e** Bar graph depicting the TGN46 or B4GALT1 degradation efficiency calculated from (**d**). Data from 3 biological replicates. One-way ANOVA with Tukey's multiple comparisons test. TGN46: n.s.

$p = 0.7079$, * $p = 0.0112$, $p = 0.0278$. B4GALT1: ** $p = 0.005$, **** $p < 0.0001$. **f** Violin plot showing the distribution of average PI(4)P staining intensity in the TGN. $n > 100$ cells. One-way ANOVA with Dunnett's multiple comparisons test. **** $p < 0.0001$. **g** Violin plot showing normalized filipin intensity in the TGN area (excluding lysosomal signal). Data from 3 independent experiments. Two-tailed Mann-Whitney U test. **** $p < 0.0001$. **h** Representative confocal images of Sac1-degron STBs treated for 2 h with DMSO or IAA, stained for ATP6V_0a2 (green) and ATP6V_1A (magenta). **i** Western blot analysis of cell lysate and secreted medium of Sac1-degron STB treated with DMSO or IAA for 15 h, with or without Neuraminidase A (Neu A) digestion and blotted for hCGβ. $n = 3$ independent experiments. **j** Western blot analysis of secreted medium of Sac1-degron STBs treated with DMSO, IAA or IAA for 15 h, with or without Neuraminidase A digestion. Blotted for hCGβ. $n = 3$ independent experiments.

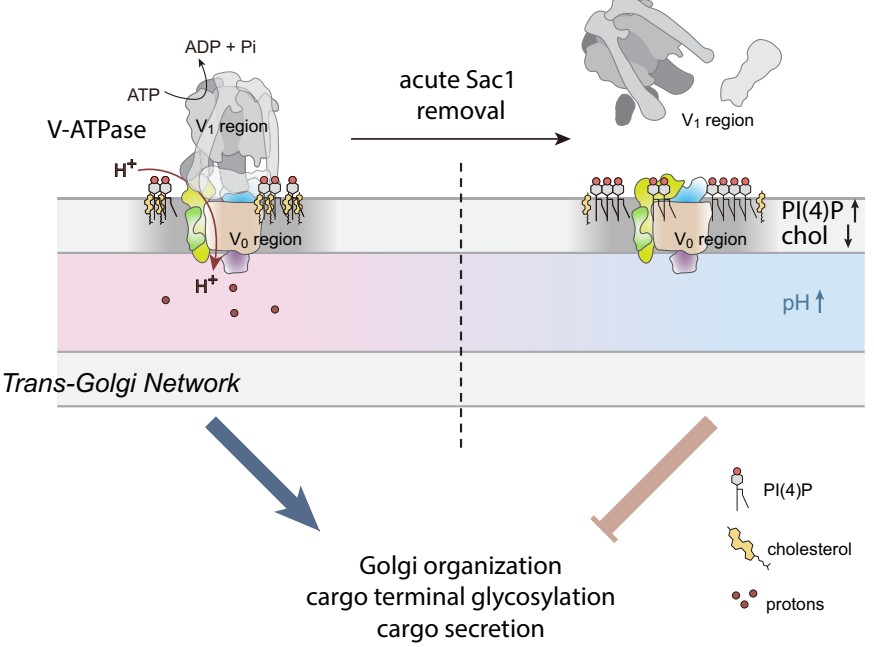

**Fig. 7 | Working model for Sac1 controlled TGN V-ATPase activity and downstream effects.** Sac1 fuelled PI(4)P/cholesterol exchange maintains the composition of the Golgi membrane, allowing V-ATPase assembly for TGN acidification, proper terminal glycosylation and cargo secretion. Rapidly after Sac1 degradation, PI(4)P is accumulated in the TGN while cholesterol is depleted. This changes the lipid microenvironment of the $V_0$ region, resulting in a conformational change in the $V_0$a2 subunit which prevents the assembly of the Golgi V-ATPase complex. This results in the deacidification of the TGN, Golgi fragmentation and defects in cargo processing and secretion.

System,s 314-BP)). The medium was replaced every 2 days and the cells routinely passaged with StemPro Accutase (ThermoFisher, A1110501) at 1:3 ratio after reaching 90-100% confluency. From day 9-10 onwards, the cells were used for experiments or further differentiated to syncytiotrophoblasts (STB). For STB differentiation, trophoblasts were dissociated with StemPro Accutase into 1-cell suspension in TB medium, counted, centrifuged at 200x*g* for 3 min and resuspended in STB medium (DMEM/F-12 supplemented with penicillin/streptomycin 100 U ml$^{-1}$ each, 0.3% fatty acid free BSA, 1% ITS-X, 0.1 mM β-Mercaptoethanol, 10 mM Forskolin (Sigma-Aldrich, F6886), 4% KnockOut Serum Replacement (Gibco, 10828028), 50 ng ml$^{-1}$ EGF, 1 μM cAMP (Sigma-Aldrich, D0260), and 5 μM ROCK inhibitor and seeded at 1x10$^5$ cells per well on Matrigel-coated 6-well plates. The medium was replaced after 3 days, and the cells were used for experiments on day 6 after STB differentiation.

**Antibodies and reagents**

Primary antibodies for western blot (WB) or immunofluorescence (IF) analysis: Rabbit polyclonal anti-TGN46 (Abcam, ab50595, IF 1:400, WB 1:1000), mouse purified anti-PI4P IgM (Echelon, Z-P004, IF 1:200), mouse monoclonal anti-GM130 (BD Biosciences, G56120-150-1, IF

1:500. WB 1:2000), rabbit polyclonal anti-GM130 (Sigma-Aldrich, G7295, IF 1:300), rabbit polyclonal anti-GRASP55 (Proteintech 10598-1-AP, WB 1:1000), polyclonal goat anti-B4GALT1 (R&D systems, AF3609, IF 1:100, WB: 1:500), anti-GFP (Abcam, ab290, WB 1:5000), mouse anti-miniIAA7 (customized from Genscript[30]), rabbit monoclonal anti-ATP6V_1A1 (Abcam, ab199326, IF 1:250, WB 1:2000), rabbit polyclonal anti-ATP6V_0d1 (ProteinTech, 18274-1-AP, WB 1:500), mouse polyclonal anti-ATP6V_0a2 (Abnova, H00023545-A01, IF 1:200), rabbit polyclonal anti-ATP6V_0a2 (Invitrogen, PA5-85284, WB 1:2000), goat polyclonal anti-LAMP2 (R&D systems, AF6228, IF 1:40), rabbit polyclonal anti-TMEM192 (Sigma-Aldrich, HPA014717, IF 1:200), DAPI (Sigma-Aldrich, D9542, IF 1:1000), rabbit polyclonal anti-hCGβ (ProteinTech, 11615-1-AP, IF 1:200), Rabbit monoclonal anti-GATA-3 (Cell Signaling Technology, #5852, IF 1:500), rabbit monoclonal anti-DAB2 (Cell Signaling Technology, #12906, IF 1:500).

Secondary antibodies for IF: Alexa Fluor™ 488 and 568 Goat anti-Mouse and anti-Rabbit IgG (Molecular Probes, A11008, A11011, A11001, A11004, 1:250), Alexa Fluor™ 647 Donkey anti-Rabbit and anti-Mouse 647 IgG (Molecular probes, A31571, A31573, 1:250), Alexa Fluor™ 488 Goat anti-Mouse IgM (ThermoFisher, A21042, 1:250). Secondary antibodies for WB analysis: HRP conjugated antibodies Goat anti-Mouse

(Bio-Rad, 1706516, 1:2000), Goat anti-Rabbit (Bio-Rad, 1706515, 1:1000) and Donkey anti-Goat (Jackson Immuno research, 705-035-147 1:5000). Reagents: cvx_IAA (final concentration 5 µM) and cvx_picoIAA (final concentration 0.5 µM)[30], DMSO (Santa Cruz, sc-358801), 1 µM Bafilomycin A1 (AppliChem c22229915), 20 nM OSW-1 (Bocsci 145075-81-6, stock 10 mM), 50 µM EN6 (Sigma-Aldrich, SML2689), 10 µM PIK93 (TargetMol, T-12454), 10 µM NC03 (Aobious, AOB17420), 4 µM Methyl-β cyclodextrin (Sigma-Aldrich, C-4555), 50 µM 8.2:1 methyl-β-cyclodextrin/cholesterol complex (Sigma-Aldrich C-4555, C-8667).

## Sac1<sup>wt</sup> and Sac1<sup>mut</sup> plasmid generation

Plasmids harboring wild-type Sac1 (Sac1<sup>wt</sup>) and a phosphatase-dead mutant Sac1 (C389S, Sac1<sup>mut</sup>) were obtained from Addgene (#108127 and #108128, respectively). The Sac1 fragments were amplified by PCR (Fw: GGAAGATCTATGGCGACGGCGGCCTA, Rev: AAAACTGCAGT-CAGTCTATCTTTTCTTTCTGGACCAGTCTGG) and cloned into the pEGFP-C1 vector. The resulting plasmids, along with pcDNA3.1 as control, were transfected into Sac1-degron HEK293A cells using X-tremeGENE™ HP DNA Transfection Reagent (Roche, 06 366 236 001). The transfection occurred overnight, after which IAA or DMSO was added for 6 h. Subsequently, cells were lysed and subjected to Western blot analysis.

## Phosphoinositide regioisomer measurement by chiral column chromatography and mass spectrometry (PRMC-MS)

A431 cells were washed twice with cold 1x PBS. Cells were scraped in 500 ml cold 1x PBS. $2.5 \times 10^6$ cells were divided over 1.5 ml tubes and centrifuged for 5 min at $1000 \times g$ at 4 °C. Cell pellets were frozen down and transferred to Tokyo Medical and Dental University (TMDU) for further processing for PRMC-MS analysis. Cellular phosphoinositide levels were measured according to[31]. Briefly, acidic phospholipids extracted from control and Sac1-depleted A431 cells were derivatized by a methylation reaction with trimethylsilyl diazomethane (Tokyo Chemical Industry, #T1146), followed by phosphoinositide regioisomer measurement by chiral chromatography and mass spectrometry. LC-MS/MS was performed using a triple quadruple mass spectrometer QTRAP6500 (ABSciex) and a Nexera X2 HPLC system (Shimadzu) combined with a PAL HTC-xt (CTC Analytics) autosampler. The obtained chromatographic data were analyzed by MultiQuant 3.0.2 software (ABSciex). Concentrations of phosphoinositide species were estimated based on the concentration of C37:4 species of the corresponding phosphoinositide regioisomers that had been added to the cell pellets as internal standards before the lipid extraction procedure.

## Lipid extraction and CAMAG TLC

Lipid extraction according to Bligh and Dyer[73]. In short, cells are washed 2x with cold 1x PBS and scraped in 2% NaCl. Subsequently, samples were homogenized using methanol (Fluka, 65543), chloroform (Merck, 34854) and ultra-pure water. Samples were centrifuged for phase separation of the lipids in the organic phase. Finally, samples were evaporated and resuspended in a chloroform/methanol (2:1) resuspension solution. 100 µg of protein was used to run a HPTLC Silica gel plate (Sigma-Aldrich, 1.05642.0001) in combination with a neutral lipid standard (1 mg ml⁻¹ FC:CE:TG) on a CAMAG Automatic TLC Sampler 4 using the visionCATS software. A running solution for neutral lipids containing 80:20:1 hexane-diethyl ether-acetic acid (Honeywell, 34859, J.T. Baker 8254, Fluka 33209) was used and the silica plate was subsequently stained with 3% CuSO₄ / 8% H₃PO₄ (Merck 1.02790.0250, Fisher Scientific, 10685082). The silica plate was charred on a stove up to 180 °C, and the plate was scanned to assess free cholesterol (FC), cholesteryl ester (CE) and triglyceride (TG) lipid content.

## Click-labeling

Quadruplicates of Sac1-degron A431 cells treated with DMSO or IAA for 2 h were labelled with 100 µM alkyne-Oleic Acid-BSA complex (0.33 mM stock complexed with BSA in 8:1 molar ratio, prepared in serum free DMEM) in CO₂ independent medium (Gibco, 18045) for 30 min. After labeling, cells were washed 1x with 1% FA-free BSA, 1x with ice-cold 1x PBS and 1x with 155 mM ammonium acetate. Lipids were extracted with a chloroform:methanol (6:1) mix with internal standards (D8-Click IS, 13C3 IS and OMICS Click IS[74]) and sonicated for 60 seconds. Chloroform and 1% acetic acid were added, and samples were vortexed and centrifuged. Lower phase was evaporated and analyzed by Mass Spectrometry[75].

## PI(4)P staining

Cells grown on coverslip were fixed with 2% paraformaldehyde (Electron Microscopy Sciences, 15710) in 1x PBS for 15 min at RT. Subsequently, cells were permeabilized with 20 µM digitonin (Sigma-Aldrich, 11024-24-1) for 5 min and blocked for 1 h in 1% fatty acid free BSA in 1x PBS (Sigma-Aldrich, A3803). Cells were incubated for 2 h at 37 °C with mouse anti-PI(4)P IgM antibody, washed 3x with 1x PBS and incubated with Alexa Fluor™ 488 Goat anti-Mouse IgM secondary antibody. Coverslips were mounted on microscopy slides using Prolong glass antifade reagent (Invitrogen, P36980).

## mCherry-P4M-SidM live-imaging

Cells were transfected with mCherry-P4M-SidM using X-tremeGENE HP DNA transfection reagent according to manufacturer's protocol. The day after transfection, live-cell imaging was performed on Zeiss LSM 880 confocal microscopy at 37 °C and 5% CO₂. The grayscale images were visualized using ImageJ software with the 'Physics' lookup table applied to enhance contrast and highlight specific intensity ranges.

## ALOD4 plasma membrane staining

Recombinant mNeonGreen-ALOD4 was a gift from Arun Radhakrishnan (UT Southwestern). For plasma membrane ALOD4 staining, cells were grown on ibidi 15 µ-slide 8-well chamber slides (Ibidi, 80826) and stained with 0.5 µM mNeonGreen-ALOD4 (diluted in 1% fatty acid free BSA in cold 1x PBS) for 30 min on ice. Cells were washed twice with cold 1x PBS, fixed with 4% PFA for 15 min, and quenched with 50 mM NH₄Cl in 1x PBS for 10 min at RT.

## Filipin staining

For filipin staining, cells were grown on glass coverslips and fixed with 4% PFA. Cells were washed 2x with 1x PBS and incubated for 30 min at 37 °C with 0.5 mg ml⁻¹ filipin (Sigma-Aldrich, F9765) diluted in 1x PBS. Cells were washed 3x with 1x PBS and mounted on microscopy slides with Mowiol-DABCO (CalBioChem, 475904, Sigma D-2522). For better intracellular visibility of filipin staining, cells were treated for 30 min with 10 mM methyl-β-cyclodextrin at 37 °C according to[76]. Cells were incubated for 30 min at RT with 0.5 mg ml⁻¹ filipin, blocked with 1% BSA for 30 min, and subsequently incubated with primary and secondary antibodies. After antibody staining, the cells were re-incubated with 0.5 mg ml⁻¹ filipin for 30 min at RT and mounted on microscopy slides with Mowiol-DABCO.

## Immunofluorescence stainings

Cells grown on glass coverslips were fixed for 15 min at RT with 4% PFA (Sigma-Aldrich, P-441244) or 10 min with ice-cold methanol, permeabilized for 10 min with 0.1 – 0.2% Triton X-100 (Sigma-Aldrich, t-8787) in 1x PBS and blocked for 30 min with 1 to 3 % BSA (Sigma-Aldrich, A-3803) in 1x PBS. Cells were incubated with primary antibodies diluted in blocking buffer for 1 to 2 h at RT or 37 °C, followed by secondary antibody incubation for 1 h at RT. Coverslips were mounted on microscopy slides using Prolong glass antifade reagent (Invitrogen, P36980).

## Image acquisition

Imaging was performed at the Biomedicum Imaging Unit at the University of Helsinki, supported by the Helsinki Institute of Life Science

(HiLIFE) and Biocenter Finland. For confocal imaging, a Zeiss LSM 880 inverted confocal microscope with Airyscan detector, 63x/1.40 PL Apo oil objective, Diode and Argon lasers was used. For FLIM and confocal imaging a Leica Stellaris 8 Falcon / DLS inverted confocal microscope, equipped with HyD detectors, 405 nm diode laser, and super-continuum laser, with 63x/1.40 HC PL APO CS2 oil or 63x/1.20 HC PL APO CS2 water objective was used. For widefield imaging a NIKON Eclipse Ti-E inverted wide-field microscope, equipped with Led-DAPI-A, TxRed-4040C, LED-Cy5-A, LED-FITC-A filter cubes, Hamamatsu Orca Flash 4.0 V2 B&W camera and Lumencor Spectra X light engine, with 40x/0.75 Plan Fluor dry objective, 60x/1.40 Plan Apo Oil or 100x/1.40 Plan Apo VC oil objective with Perfect Focus System was used. A Leica Upright DM6 B epifluorescence microscope was used for immuno-fluorescent staining of TB differentiation markers.

### TGN pH measurements using fluorescence lifetime microscopy (FLIM)

FLIM pH measurements were performed according to[34]. Briefly, Sac1-degron cells were transfected with GalT-RpHLuorin2, for *trans*-Golgi expression of pHLuorin2, (Addgene, #171719) using X-tremeGENE HP DNA transfection reagent according to manufacturer's protocol. 5 h after transfection cells were seeded onto an 8-well Lab-Tek chambered cover glass (Thermo Fisher, 155411) coated with 5 μg ml$^{-1}$ fibronectin (Roche, 11051407001). FLIM imaging was performed on a Leica Stellaris 8 Falcon / DLS with an OKO lab full-enclosure incubator set at 37 °C, equipped with a 63x/1.20 HC PL APO CS2 water objective. GalT-pHLuorin2 was excited at 488 nm with a pulsed white light laser at 80 mHz. An HyD X detector was set at 502-530 nm, and photons were collected for 1 min. Lifetime histograms were fitted with a mono-exponential decay function. pH calibrations were performed before every experiment using an Intracellular pH Calibration Buffer kit (Thermo Fisher Scientific, P35379) to generate a standard curve to determine pH levels.

### Transmission electron microscopy

Cells grown on cover slips (thickness #1) were fixed with 2.5% glutar-aldehyde (EM-grade, Sigma-Aldrich, G-7651) in 0.1 M sodium cacody-late buffer, pH 7.4, and flat embedded according to[77]. Briefly, after 30 min fixation at RT, the cells were post-fixed with reduced 1% osmium tetroxide for 1 h on ice, gradually dehydrated in ethanol and acetone, and infiltrated into epoxy (TAAB 812 resin, T030). Sections ca. at height of Golgi were cut from the polymerized sample blocks, picked-up on single slot grids, and post-stained with uranyl acetate and lead citrate. TEM imaging was performed using Hitachi HT7800 microscope (Hitachi High-Technologies, Tokyo, Japan) operated at 100 kV. For unbiased analysis 10 cells were selected at low magnifica-tion and montage images were acquired covering the whole Golgi area at nominal magnification of 6000x (pixel size 1.84 nm) using a bottom mounted Rio9 CMOS-camera (AMETEK Gatan Inc., Pleasanton, CA) and SerialEM software (https://bio3d.colorado.edu/SerialEM, version 3.8.1). The analysis of the Golgi morphology was done using pseudo-nymized TEM micrographs (MIB software[78]).

### Image quantification

Whole cell area, Golgi region or nuclei were segmented with interactive machine learning algorithms based software Ilastik (version 1.3.2)[79]. Cellular PI4P (Supplementary Fig. 4b), ALOD4 (Supplementary Fig. 4d), Filipin (Supplementary Fig. 4f) and TGN46 (Fig. 3e and Supplementary Fig. 5b), Golgi PI4P (Fig. 4a and Supplementary Fig. 7e) and cholesterol (Fig. 4c and Supplementary Fig. 7f), lysosomal cholesterol (Supple-mentary Figs. 4i and 7f), and Western blot intensities were measured using Fiji ImageJ (version 1.54 f). For colocalization analysis, binary images of ATP6V$_1$A and ATP6V$_0$a2 subunits (Figs. 3h and 4g) were obtained with Ilastik. Pearson's correlation coefficient was calculated using the command 'colocalization' in Fiji ImageJ. The co-localization

of ATP6V$_1$A and ATP6V$_0$a2 with the lysosomal marker LAMP2 and the Golgi marker GM130 was assessed in a similar way (Fig. 3j, l).

### Quantitative proteomic analysis

**Mass spectrometry.** Mass spectrometry analysis was performed at the Turku Proteomics Facility, University of Turku and Åbo Akademi Uni-versity, supported by Biocenter Finland. Quantitative proteomic ana-lysis of 3 h DMSO and 5 μM cvxIAA treated A431 Sac1-degron cells was performed using biological triplicates. Cell samples were lysed, and proteins were denatured with 8 M urea in 50 mM Tris-HCl, pH 8. Samples were reduced with 10 mM D,L-dithiothreitol and alkylated with 40 mM iodoacetamide. Samples were digested overnight with sequencing grade modified trypsin (Promega). After digestion, pep-tides were desalted with a SepPak C18 96-well plate (Waters), evapo-rated to dryness and stored at −20 °C. Digested peptides were dissolved in 0.1% formic acid and peptide concentration was deter-mined with a NanoDrop device (Thermo Fisher Scientific). The Liquid Chromatography Electrospray Ionization Tandem Mass Spectrometric (LC-ESI-MS/MS) analysis was performed on a nanoflow HPLC system (Thermo Fisher Scientific, Easy-nLC1200) coupled to the Orbitrap Fusion Lumos mass spectrometer (Thermo Fisher Scientific) equipped with a nano-electrospray ionization source. 800 ng peptides samples were loaded and separated inline on a 25 cm C18 column (75 μm x 25 cm, ReproSil-Pur 1.9 μm 120 Å C18-AQ, Dr. Maisch HPLC GmbH, Ammerbuch-Entringen, Germany) heated to 60 °C. The mobile phase consisted of water with 0.1% formic acid (solvent A) or acetonitrile/water (80:20 (v/v)) with 0.1% formic acid (solvent B). A 170 min gra-dient was used to elute peptides (95 min from 5% to 21% solvent B followed by 75 min from 21% to 36% solvent B).

Samples were analyzed by a data independent acquisition (DIA) method. MS data was acquired automatically by using Thermo Xcali-bur 4.2 software (Thermo Fisher Scientific). In a FAIMS-DIA method a duty cycle contained two compensation voltages (−50V and −70V) with one full scan (400 −1000 m/z) and 30 DIA MS/MS scans covering the mass range 400 – 1000 with variable width isolation windows in each of the compensation voltages.

**Data processing parameters.** Data analysis consisted of protein identifications and label free quantifications of protein abundances. Data was analyzed by Spectronaut software (Biognosys, version 15.6.2). DirectDIA approach was used to identify proteins and label-free quantifications were performed with MaxLFQ.

Main data analysis parameters in Spectronaut:
- Enzyme: Trypsin/P
- Missed cleavages: 2
- Fixed modifications: Carbamidomethyl (Peptides were alkylated with iodoacetamide)
- Variable modifications: Acetyl (protein N-term) and oxidation (M)
- Protein database: Swiss-Prot 2021_04 Homo sapiens
- Precursor FDR Cutoff: 0.01
- Protein FDR Cutoff: 0.01
- Quantification MS level: MS2
- Quantification type: Area under the curve within integration boundaries for each targeted ion
- Normalization: Local normalization (Based on RT dependent local regression model described by[80])
- Differential abundance analysis: Unpaired Student's t-test with combined MS1 + MS2 statistical model[81] and multiple-testing correc-tion of the p-values with Benjamini-Hochberg method (q-values).

**Gene Ontology (GO) Enrichment Analysis.** GO Enrichment Analysis were performed online (https://geneontology.org/docs/go-enrichment-analysis/)[82,83]. This service connects to the analysis tool from the PAN-THER Classification System, which is kept up to date with GO annotations[84]. In brief, lists of significantly upregulated (p < 0.05,

FC > 1.33 (cvxIAA 4 h/DMSO 4 h)) and downregulated (p < 0.05, FC < 0.75 (cvxIAA 4 h/DMSO 4 h)) proteins were subjected to GO enrichment analysis and categorized into two main GO categories: biological process (BP) and cellular component (CC). As a reference, the list of all genes of Homo sapiens was used. Fisher's Exact Test was employed to identify significant enrichments with the False Discovery Rate (FDR) correction method to adjust p-value for multiple testing. FDR < 0.05 was set as significance threshold.

## Western blotting

Cells were washed twice in 1x PBS and lysed in cold RIPA lysis buffer (1% NP-40, 0.1% SDS, 0.5% Sodium Deoxycholate in 1x TBS) with freshly added protease inhibitors (25 µg ml$^{-1}$ chymostatin (Sigma-Aldrich, C-7268), 25 µg ml$^{-1}$ leupeptin (Sigma-Aldrich, L-2023), 25 µg ml$^{-1}$ antipain hydrochloride (Sigma-Aldrich, A-6191) and 25 µg ml$^{-1}$ pepstatin A (Sigma-Aldrich, P-5318)). DC$^{TM}$ protein Assay Kit I (Bio-Rad, 5000111) was used to determine protein concentration. 5x Sample buffer containing β-Mercaptoethanol was added to the cell lysates to a final dilution of 1x and lysates were denatured for 1 h at 37 °C. 10 – 20 µg of protein was loaded on Mini-Protein TGX Stain-Free gels (Bio-Rad, 1610181, 1610183, 1610185) and run at 120 V in 1x SDS-Page Running buffer. Pageruler Plus prestained protein ladder (Thermo Fisher, 26619) was used. Gels were activated with the ChemiDoc MD Imaging System (Bio-Rad) and transferred to a 0.45 µm Low Fluorescence PVDF membrane (Bio-Rad, 1704274). Membranes were blocked for 45 min at RT with 5% skim milk (Valio) in 0.1% Tween-20 (Fisher Scientific, BP337) in TBS (TBS-t), and subsequently incubated with the primary antibody overnight at 4 °C. Membranes were washed 3x with TBS-t and incubated for 45 min at RT with an HRP-conjugated secondary antibody. Membranes were washed 3x with TBS-t, incubated for 1 min with Clarity Western ECL substrate (Bio-Rad, 1705061) and imaged using a ChemiDoc MD Imaging system (BioRad). Uncropped and unprocessed blots are provided in the Source Data.

## RT-qPCR

Total RNA was extracted using the NucleoSpin RNA Isolation Kit (Macherey-Nagel, Cat. 740955-250). Subsequently, 1 µg of RNA was reverse transcribed into cDNA according to the SuperScript VILO cDNA Synthesis Kit (Invitrogen, Cat. 11754050). RT-qPCR was then performed on a LightCycler 480 II system (Roche) using LightCycler 480 SYBR Green I Master Mix (Roche, 04707516001). The amplification protocol consisted of an initial denaturation at 95 °C for 5 minutes, followed by 40 cycles with the following conditions: 95 °C for 10 s, 64 °C for 15 s, and 72 °C for 20 s. The expression levels of target mRNAs were quantified relative to 18S ribosomal RNA using 2$^{-\Delta\Delta CT}$ method.

Primer pairs were designed for the amplification of the genes of interest. Specifically, TGN46: AGCTGGAGTACGGCCTTCT (fw) and GATGCGACTTGGTAGAGCCTC (rev); B4GALT1: GTATTTTGGAGGTG TCTCTGCTC (fw) and GGGCGAGATATAGACATGCCTC (rev); 18S rRNA: CGGCTACCACATCCAAGGAA (fw) and GCTGGAATTACCG CGGCT (rev).

## Subcellular fractionations

**Membrane/cytosol fractionation.** To investigate membrane association of the ATP6V$_1$A subunit, A431 cells were scraped from 10-cm dishes and transferred to 1.5 ml tubes. Cell pellets were centrifuged at 800x*g* for 3 min at 4 °C and washed with cold 1x PBS to remove residual culture medium. Cells were re-suspended into 500 µl fractionation buffer (20 mM HEPES pH 7.4, 10 mM KCl, 2 mM MgCl$_2$, 1 mM EDTA, 1 mM EGTA, 1 mM DTT and protease inhibitors) and passed 100x through a 26-gauge needle using a 1 ml syringe. 5 µl cell homogenate was observed with inverted phase contrast microscope (Olympus) to ensure over 90% cells were broken. Pellets containing nuclei were removed by 5 min 1,000x*g* centrifugation at 4 °C. The post-nuclear

supernatant (PNS) was subjected to 100,000x*g* ultracentrifugation for 1 h at 4 °C to obtain organellar membrane fraction pellet. Membrane fraction was dissolved in 2x SDS-PAGE sample buffer. Supernatant containing cytosolic proteins was mixed with equal amount of 20% cold trichloroacetic acid (TCA) and placed on ice for 2 h. After 15 min, samples were centrifuged at 16,200x*g* at 4 °C, and denatured protein sediment was collected and washed with cold acetone twice. Protein sediment was dissolved in 2x SDS-PAGE sample buffer. ATP6V$_1$A levels in the membrane and cytosol were analyzed by Western blotting.

**Sucrose density gradient fractionation.** To investigate the membrane lipid environment surrounding ATP6V$_0$a2 subunit, cells were scraped from 10-cm dishes and transferred to 1.5 ml tubes. Cell pellets were centrifuged at 800x*g* for 3 min at 4 °C and washed with cold 1x PBS to remove residual culture medium. Cells were re-suspended into 600 µl hypotonic buffer (10 mM HEPES pH 7.4, 0.3 M sucrose, protease inhibitors) and passed 100x through a 26-gauge needle using a 1 ml syringe. 5 ml cell homogenate was observed with inverted phase contrast microscope (Olympus) to ensure over 90% cells were broken. Pellets containing nuclei were removed by centrifugation for 5 min at 1,000x*g* at 4 °C. 500 µl PNS was mixed with 3 ml 2.5 M sucrose (10 mM HEPES pH 7.4) and placed on the bottom of the tube (Beckman, 331374). Then, 1 ml 1.9 M, 1 ml 1.4 M, 2 ml 1.2 M 3 ml 1.0 M and 1.5 ml 0.3 M sucrose solutions in 10 mM HEPES pH 7.4 were placed above PNS mixture, sequentially from bottom to the top to create a sucrose gradient. Samples were ultracentrifuged in a Beckman SW40 Ti rotor at 264,000x*g* for 16 h. Afterwards, twelve fractions of 1 ml were collected from the top and mixed with equal amounts of 20% cold TCA and placed on ice for 2 h. After centrifugation for 15 min at 16,200x*g* at 4 °C, denatured protein sediment was collected and washed with cold acetone twice. Protein sediment was dissolved in 2x SDS-PAGE sample buffer. Distribution of ATP6V$_0$a2 across the fractions was analyzed by Western blotting.

## PNGaseF and neuraminidase A digestions

PNGaseF (NEB, P0704) digestions were performed according to manufacturer's protocol. Sac1-degron cells were lysed in RIPA lysis buffer (1% NP-40, 0,1% SDS, 0.5% Sodium Deoxycholate, in 1x TBS). After protein concentration determination using the DC$^{TM}$ Protein Assay Kit I (Bio-Rad, 5000111), 10 µg of glycoprotein was mixed with 2 µl of 10x Glycobuffer. H$_2$O was added to make a 20 µl total reaction volume. Finally, 2 µl of PNGaseF was added and samples were incubated for 16 h at 37 °C.

α2-3,6,8,9 Neuraminidase A (NEB, P0722) reaction was performed according to manufacturer's protocol. In short, 2 µg of glycoprotein were mixed with H$_2$O to make a total reaction volume of 18 µl. 2 µl of 10x GlycoBuffer 1 was added, followed by 2 µl of α2-3,6,8,9 Neuraminidase A. The reaction mix was incubated at 37 °C for 1 h.

5x sample buffer was added to the samples and the samples were analyzed on a 7.5% Mini-Protein TGX Stain-Free gels (Bio-Rad, 1610181). A similar amount of undigested sample was loaded on the same gel to determine PNGaseF and α2-3,6,8,9 Neuraminidase A activity.

## Analysis of secreted proteins

Sac1-degron TBs were seeded on 10 cm dishes. Cells were washed 3x with 1x PBS and medium was changed to serum-free medium containing DMSO or IAA with or without EN6. After 15 h medium was collected and centrifuged for 10 min at 2,300x*g*. Meanwhile, cells were lysed according to western blot protocol. After centrifugation, 5x SDS-PAGE sample buffer was added to secreted medium, and samples were denatured for 10 min at 95 °C and analyzed by Western blotting.

## gnomAD lof.pLI analysis

From the Genome Aggregation Database (gnomAD) dataset v4.0.0 (gnomAD (broadinstitute.org)), the publicly available genetic

constraint metrics TSV data was downloaded containing the lof.pLI score for genes, indicating the probability of a gene of being loss-of-function intolerant. A probability of 0 indicates a gene is highly tolerant to loss-of-function mutations, while a lof.pLI 1 indicates that a gene does not tolerate loss-of-function mutations. Data was plotted as a density plot using RStudio highlighting the lof.pLI score of *SACM1L*.

## Atomistic molecular dynamics simulations

The simulation model is based on the $V_O$ region hosted in a lipid membrane. As the initial structure of $V_O$, we used the PDB ID: 6WM2[40]. The cryo-EM structure includes the lysosomal $V_O$a1 subunit that was replaced by the $V_O$a2 subunit structure given by the AlphaFold database entry Q9Y487[85,86]. Using CHARMM-GUI[87], the modified $V_O$ structure was then inserted in a 1-oleoyl-2-palmitoyl-glycero-3-phosphocholine (POPC) membrane, with varied concentrations of phosphatidylinositol 4-phosphate (PI(4)P) and cholesterol (Table 1). Membranes were comprised of a total number of 980 lipids that were symmetrically distributed in the two leaflets. In additional simulations, an asymmetric distribution of PI(4)P was also explored but was observed not to affect conclusions. Further, while a number of additional mixtures of POPC, PI(4)P and cholesterol were investigated, they did not bring forth additional insight. Hence, for the sake of clarity, the focus here is on the systems in Table 1. The membrane complex was solvated in a solution of counterions to neutralize the system, with 150 mM KCl. The CHARMM36m force field[88] was used for the protein and lipids, the TIP3P model[89] for water molecules, and a compatible parameter set for the ions[90].

The models were simulated using the GROMACS 2022.3 engine[91] using the Verlet scheme and the hydrogen mass repartitioning method[92] to achieve a 4 fs timestep. Periodic boundary conditions were applied in all three dimensions. All production simulations were performed in the NpT ensemble. The Nosé-Hoover thermostat[93,94] was used to maintain the temperature at 310 K with the protein, the membrane, and the solvent (water and KCl) coupled to separate temperature baths with a time constant of 1.0 ps. The Parrinello-Rahman barostat[95] was used for semi-isotropic pressure coupling at 1 atm with a time constant of 5 ps and a compressibility value of $4.5 \times 10^{-5}$ bar$^{-1}$. The LINCS algorithm[96] was used to constrain all bonds. For electrostatic interactions, a real space cut-off of 1.2 nm was used. Long-range electrostatic interactions were computed using the Particle-Mesh Ewald method[97] with a Fourier spacing of 0.12 nm and a fourth-order interpolation. For the van der Waals interactions, a Lennard-Jones potential with a force-switch between 1.0 and 1.2 nm was used.

All visualizations were implemented with VMD[98]. The analysis of the simulated data (except for deep learning-based analyses) was carried out using the MDAnalysis python package[99,100].

## Autoencoder-based dimensionality reduction

To detect conformational states in the simulated protein data, the dimensionality of the raw configurational data was reduced using a denoising autoencoder architecture[101] as implemented in PyTorch[102]. The L1-loss (Mean Absolute Error loss) function was used for assessing the reconstruction quality. Only the C-alpha atoms of the protein structure were used for the dimensionality reduction. The encoded 2D space was mapped on a 3 × 3 grid. Each grid point acted as a cluster forming a region with the constraint that the Euclidean similarity of the input data in its immediate vicinity was maximized using the L2-loss (Mean Squared Error Loss) function. This additional loss function assisted in creating preformed clusters of highly similar structures. The model was trained on only half of the (shuffled) simulation frames to avoid overfitting. The reconstruction error was then evaluated on the remaining holdout data. To make the data rotationally and translationally invariant, the total rotations and translational movement was removed from the entire data set by fitting it on a reference structure.

The model was further regularized with a 20% dropout rate on each layer. After manually testing a wide variety of architectures, an autoencoder with two hidden layers in the encoder module and two hidden layers in the decoder module was finally chosen. The encoder and decoder modules had a mirrored structure with the former module with 1024 neurons in the first hidden layer, and a reversed arrangement in the latter module. The encoded layer was created with only two neurons to obtain a 2D reduced dimensionality. The non-linearity in encoding was provided by the ReLU[103] activation function. The model was then trained for 500 epochs using a fixed seed with the intent of maintaining reproducibility. At the end of training, a mean absolute reconstruction error (MAE) of 1.01 Å in the holdout data was obtained. This model was then saved and used for further analysis.

## Gaussian Mixture model-based clustering

To find clusters in the dimensionality-reduced space obtained from the autoencoder, a Bayesian Gaussian Mixture Model (BGMM) as implemented in the scikit-learn package[104] was used. A full covariance matrix for the 2D Gaussians was used to allow for flexible fitting of the Gaussian density on the data. The number of clusters present in the data was selected to be 4, based on first using a large value for this parameter and allowing the BGMM to regularize the number. The clusters are divided by their decision boundaries, which represent the region where the uncertainty of a cluster label approaches 50%.

## Partial least squares-based discriminant analysis

The functional difference between the assembly-promoting cluster 2 and the other three clusters is clearly revealed through Partial Least Squares-based discriminant analysis (PLS-DA). This was done by creating a vector whose length corresponds to the total number of structures in all simulations and setting the value of this vector to 1 for structures classified in cluster 2, and 0 for structures classified in other clusters. The Scikit-learn PLS package was then used to determine whether it was possible to create a linear model that would separate cluster 2 from the other clusters. Half of the shuffled data was used to train this model. The rest of the data was left for model validation. A six-component model was found to have a robust ability to predict the labels in the validation set with $R^2 > 0.75$. This model was then visualized by linear interpolation between values 0 and 1 to interpret the differences between the clusters 1, 3, and 4 at one end, and the cluster 2 at the other end.

## Statistical analysis

Statistical analyses were performed and plotted using Graphpad Prism (version 9.4.1). Data were checked for normal distribution. For comparisons of two groups, Student's t-test was used. For comparisons of two or more groups, a One-way ANOVA with multiple comparisons test was used.

## Reporting summary

Further information on research design is available in the Nature Portfolio Reporting Summary linked to this article.

# Data availability

The gnomAD constraint metrics TSV data set v4.0.0 is publicly available via: gnomAD (broadinstitute.org). The mass spectrometry proteomics data have been deposited to the ProteomeXchange Consortium via the PRIDE[105] partner repository with the dataset identifier PDX065794. Source data are provided with this paper.

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

## Acknowledgements

We thank the HiLIFE and Biocenter Finland supported Biomedicum Imaging and Electron Microscopy Units and Turku Proteomics Facility for infrastructure support, Giray Enkavi and Waldemar Kulig for discussions on molecular dynamics simulations, Ábel Szkalisity for support on data analysis and the generation of the interactive volcano plot, and Anna Uro for technical assistance. X.Z. was supported by Academy of Finland grant 332096, M.M.v.d.S. and H.L. received funding from the European Union's Horizon 2020 Research and Innovation programmes under the Marie Sklodowska-Curie Personal Fellowship (M.M.v.d.S.) no. 101059424 and the Marie Sklodowska-Curie Grant Agreement (H.L.) no. 953489. E.I. was supported by the Academy of Finland (grant no. 324929), Foundation Leducq 19CVD04, Jane and Aatos Erkko Foundation and Sigrid Jusélius Foundation, T.S. was supported by AMED (grant no. 24gm1710007), by TMDU under Multilayered Stress Diseases (JPMXP1323015483) and Medical Research Center Initiative for High Depth Omics. T.S. was partly supported by AMED-CREST (JP25gm1710001h0003), Nanken-Kyoten, The Multilayered Stress Diseases program and the Initiative for High Depth Omics, IST. I.V. thanks the Helsinki Institute of Life Science (HiLIFE) Fellow program, Sigrid Jusélius Foundation, Lundbeck Foundation, and Academy of Finland (project IDs: 335527, 331349, 336234, 364185) for financial support. The authors thank CSC–IT Center for Science Ltd (Espoo, Finland) for providing computational resources.

## Author contributions

E.I. conceived and oversaw the project. X.Z., M.M.v.d.S., H.L. and M.H. performed experiments and analyzed data. S.L. provided Sac1-degron cell lines and technical support for cell engineering. H.V. and E.J. performed electron microscopic analysis. C.T. performed click-lipid MS experiments. O.P. performed gnomAD dataset analysis. S.M., J.S. and T.S. performed PRMC-MS experiments and analysis. S.K. and I.V. performed and analyzed atomistic molecular dynamics simulations. X.Z., M.M.v.d.S. and E.I. wrote the manuscript, and all authors edited the manuscript.

## Competing interests

The authors declare no competing interests.
