## [Transparent Peer Review file · Nature Communications]

Control of Golgi- V-ATPase through Sac1-dependent co-regulation of PI(4)P and cholesterol

Corresponding Author: Professor Elina Ikonen

Version 0:

Reviewer comments:

Reviewer #1

(Remarks to the Author)

I carefully went through the revised version that I previously reviewed. I think that the authors have done an excellent job in revising this manuscript and thank them for all the work they invested. Although they did not identify the mechanism of TGN46 degradation, I agree with their verdict that the precise mechanism is probably outside the scope of the current study. It is up to the authors and the Editor to decide whether the Figure with ATG7-KO cells might not be useful in the supplement of the paper. The microscopy figures are now much more convincing. I have no further comments or concerns and think that the work should be published.

Reviewer #2

(Remarks to the Author)

In the revised manuscript, Zhou, van der Stoel, and colleagues have addressed many of the concerns raised in the earlier version. Notably, they have extensively discussed prior studies on Sac1 and OSBP, clarifying the knowledge gap in the field. Furthermore, they have provided novel insights into the potential mechanisms by which PI4P and cholesterol regulate the formation of the Golgi V-ATPase complex using molecular dynamic simulations.

While the current manuscript has been improved, there remain issues that the authors should address prior to the publication.

1. The representative active conformation shown in Figure 5J suggests the importance of the salt bridges in the molecular dynamics simulations. Could the authors perform additional experiments, such as mutagenesis studies, to validate these findings?
2. As authors discussed in the discussion section, the changes in the distribution of PI4P/Cholesterol, following acute depletion of Sac1, are indeed consistent with disruption of OSBP-mediated counter exchange of PI4P and Cholesterol. While Sac1 most likely affects a variety of other lipid transfer proteins beyond OSBP, as the authors discussed, authors showed in the Reviewer Fig. 2 that OSW-1 treatment induces TGN46 degradation, phenocopying the impact of acute Sac1 depletion. The authors should include this data in the manuscript as supportive evidence that OSBP inhibition phenocopies Sac1 depletion. In the model depicted in Figure 7, it is helpful to indicate that Sac1 removal results in increase in PI4P and decrease in cholesterol of the TGN, likely due to OSBP inhibition (as indirect consequence). In the legend, it can be stated "Sac1 fueled PI(4)P/Cholesterol exchange, likely mediated by OSBP, etc." for clarity.
3. On page 7, line 175, the manuscript states, "in line with increased cholesterol delivery to the ER." However, no supporting evidence is provided for this claim. It is more likely due to reduced cholesterol delivery to the TGN from the ER, leading to cholesterol accumulation in the ER.

Reviewer #3

(Remarks to the Author)

Zhou et al. have examined the cellular consequences of acute depletion of PI(4)P phosphatase, Sac1, from cultured human cells. The authors finding that depletion of Sac1 triggered a significant increase in PI(4)P levels and decrease in cholesterol levels in the TGN. These changes were followed by TGN fragmentation, loss of protein glycosylation and secretion. The authors contend that the primary stimulus for these changes in TGN morphology is the loss of TGN acidification stemming from disassembly of the vacuolar ATPase complex. On the whole the manuscript is well-written and described the interesting consequences stemming from the acute depletion of Sac1.

Several concerns were raised in the prior round of reviews, that have for the most part, addressed in the revised manuscript. The manuscript is now suitable for publication in Nature Communications.

Reviewer #4

(Remarks to the Author)

Zhou et al. proposed critical role of the phosphoinositide phosphatase Sac1 in regulating Golgi morphology and function through its impact on lipid composition, specifically PI(4)P and cholesterol levels. The study utilizes an auxin-inducible degradation system to rapidly deplete Sac1 from human cells, leading to a spike in PI(4)P and subsequent decrease in cholesterol levels within the trans-Golgi network (TGN). This shift results in TGN disintegration and impaired glycosylation processes, significantly affecting protein processing such as human chorionic gonadotropin (hCG) secretion. The findings highlight the Golgi V-ATPase as a key effector controlled by lipid exchanges driven by Sac1 activity, revealing insights into how lipid composition influences organelle integrity and cellular functions.

The manuscript presents a compelling investigation into the role of Sac1 in regulating Golgi morphology and function through lipid composition, specifically focusing on PI(4)P and cholesterol dynamics. The use of an auxin-inducible degradation (AID) system to study acute effects following Sac1 depletion is innovative and allows for clear temporal analysis. The findings are significant in elucidating how lipid exchanges influence V-ATPase assembly and Golgi integrity. However, there are several areas where clarity could be improved, as well as opportunities to strengthen the discussion regarding broader implications as follow.

1. It would be good for reader with broader area of expertise if you could provide more context about previous studies about Sac1 functionality beyond its role as a phosphatase. Moreover, it would also be beneficial to include a brief overview of how dysregulation in these pathways may contribute to diseases associated with glycosylation defects.

2. In my opinion, to present the quantitative data from proteomics, consider using graphical representations alongside tables for easier interpretation.

3. Strengthen discussions around observed changes in other phosphoinositide besides PI(4)P, even if they didn't show significant changes; any insights might foster deeper understanding.

4. Provide a more robust connection between laboratory findings and clinical implications; this will enrich readers' understanding regarding how these molecular interactions translate into developmental processes or disease states.

5. Discuss potential feedback mechanisms between lipids that could affect cellular homeostasis beyond just direct perturbations caused by Sac1 loss. This could invite further exploration from future studies.

6. Detail of the setup parameters such as force fields used are crucial aspects when discussing simulation reliability. Please clearly specify the reason of the force fields utilized for the lipid bilayers and protein complexes. In your case, you have used CHARMM36m force fields for both proteins and lipid bilayer which may be too generalized because there are several well-known force fields separately designed for protein and lipids.

7. The MD simulations revealed that variations in lipid composition significantly affect the conformational states of the V0 region of the Golgi V-ATPase. Specifically, higher levels of PI(4)P led to conformations that disrupt V0-V1 complex assembly by destabilizing critical interactions between subunits. Conversely, introducing cholesterol into PI(4)P-rich membranes tend to promoted assembly of membrane proteins like V-ATPase by stabilizing certain conformational states conducive to binding with the intermembrane region. This underscores cholesterol's role as an essential modulator in maintaining protein complexes via lipid environment regulation. Therefore, the author should discuss more detail about their findings by provide a more quantitative comparison of how varying lipid compositions impacted V0 region conformations.

Version 1:

Reviewer comments:

Reviewer #1

(Remarks to the Author)

The authors have further improved their work and I have no further comments for improvement. I think that they responded well to all comments by me and the other reviewers.

Reviewer #2

(Remarks to the Author)

The authors have addressed all my concerns, and the manuscript has been much improved. I have no further comments or concerns, and I believe the work should be published.

Reviewer #3

(Remarks to the Author)

The concerns raised in the previous rounds of review have been satisfactorily addressed by the authors. The manuscript is now suitable for publication without needing additional changes.

Reviewer #4

(Remarks to the Author)

I have no further comments and suggestions for the authors. They have done an excellent job to provide more insight based on their result and improve their discussion. I can recommend for acceptance of the article at this stage.

Point-by-point responses to the remaining comments of the Reviewers:

Reviewer #1 (Remarks to the Author):

I carefully went through the revised version that I previously reviewed. I think that the authors have done an excellent job in revising this manuscript and thank them for all the work they invested. Although they did not identify the mechanism of TGN46 degradation, I agree with their verdict that the precise mechanism is probably outside the scope of the current study. It is up to the authors and the Editor to decide whether the Figure with ATG7-KO cells might not be useful in the supplement of the paper. The microscopy figures are now much more convincing. I have no further comments or concerns and think that the work should be published.

We thank the Reviewer for the positive remarks and for helping us improve this study. We felt that providing the negative data on ATG7KO cells (included in our previous Response to the Reviewer, i.e. that the mechanism of TGN46 degradation is not autophagic in nature but apparently follows a novel, non-canonical degradation pathway that remains to be characterized) would perhaps rather distract the reader from the main focus of the present study, and therefore decided to leave it out.

Reviewer #2 (Remarks to the Author):

In the revised manuscript, Zhou, van der Stoel, and colleagues have addressed many of the concerns raised in the earlier version. Notably, they have extensively discussed prior studies on Sac1 and OSBP, clarifying the knowledge gap in the field. Furthermore, they have provided novel insights into the potential mechanisms by which PI(4)P and cholesterol regulate the formation of the Golgi V-ATPase complex using molecular dynamic simulations.

While the current manuscript has been improved, there remain issues that the authors should address prior to the publication.

We thank the Reviewer for carefully assessing our manuscript and for the valuable feedback. We have now revised the manuscript to address the remaining issues, as described below.

1. The representative active conformation shown in Figure 5J suggests the importance of the salt bridges in the molecular dynamics simulations. Could the authors perform additional experiments, such as mutagenesis studies, to validate these findings?

We thank the Reviewer for bringing up this point. Unfortunately, validating these findings experimentally will be quite challenging as the V-ATPase is a large multi-subunit protein complex. Extopic overexpression (or rescue on top of a $V_{\alpha}2$ KO cell line) of wild-type or mutant

V_0a2 constructs may not lead to the correct assembly of this subunit with the remaining, endogenously expressed subunits of the V_0 complex in the ER. On the other hand, to our knowledge manipulations of the endogenous V_0a2 encoding locus successfully to generate point mutations has not been achieved in the available literature.

Thus, as the residues involved in the salt bridges were identified by molecular dynamics simulations and at this moment, we will not be able to validate their roles experimentally, we decided to tone down the respective results, changing the text (**line 253 – line 272**) and figures (**Fig. 5, Suppl. Fig. 6**) as follows: We moved the findings related to these salt bridges to the supplemental figures (now **Suppl. Fig. 6e, f**), and added in the text (**line 259 – 260**): ‘While these salt bridges appear most frequently in cluster 2 conformations, their role(s) in the conformational changes of the V_0a2 head remain to be elucidated.’

Please note that along with this change, we have also strengthened the paper and its conclusions with additional simulation analyses. In practice, this means that Fig. 5 was revised to include additional data regarding how PI(4)P and cholesterol affect the conformation of the V_0 region, and how they facilitate the assembly of the V_0-V_1 complex, in response to the comments of Reviewer #4.

2. As authors discussed in the discussion section, the changes in the distribution of PI(4)P/Cholesterol, following acute depletion of Sac1, are indeed consistent with disruption of OSBP-mediated counter exchange of PI(4)P and Cholesterol. While Sac1 most likely affects a variety of other lipid transfer proteins beyond OSBP, as the authors discussed, authors showed in the Reviewer Fig. 2 that OSW-1 treatment induces TGN46 degradation, phenocopying the impact of acute Sac1 depletion. The authors should include this data in the manuscript as supportive evidence that OSBP inhibition phenocopies Sac1 depletion. In the model depicted in Figure 7, it is helpful to indicate that Sac1 removal results in increase in PI(4)P and decrease in cholesterol of the TGN, likely due to OSBP inhibition (as indirect consequence). In the legend, it can be stated “Sac1 fueled PI(4)P/Cholesterol exchange, likely mediated by OSBP, etc.” for clarity.

Thank you for pointing this out. As requested, we have now included the OSW-1 treatment data in **Supplemental figure 2h, i**. and added in the text that OSW-1 phenocopies Sac1 depletion (**line 118 – 120**). Additionally, we changed the text in the Figure 7 legend stating that the increase in PI(4)P and decrease in free cholesterol levels upon acute Sac1 degradation is likely due to OSBP inhibition (**line 1264**).

3. On page 7, line 175, the manuscript states, “in line with increased cholesterol delivery to the ER.” However, no supporting evidence is provided for this claim. It is more likely due to reduced cholesterol delivery to the TGN from the ER, leading to cholesterol accumulation in the ER.

We agree with the Reviewer and have changed the text in **line 181 – 182** to: ‘likely due to reduced cholesterol delivery from the ER to the TGN, resulting in cholesterol accumulation in the ER’.

Reviewer #3 (Remarks to the Author):

Zhou et al. have examined the cellular consequences of acute depletion of PI(4)P phosphatase, Sac1, from cultured human cells. The authors finding that depletion of Sac1 triggered a significant increase in PI(4)P levels and decrease in cholesterol levels in the TGN. These changes were followed by TGN fragmentation, loss of protein glycosylation and secretion. The authors contend that the primary stimulus for these changes in TGN morphology is the loss of TGN acidification stemming from disassembly of the vacuolar ATPase complex. On the whole the manuscript is well-written and described the interesting consequences stemming from the acute depletion of Sac1.

Several concerns were raised in the prior round of reviews, that have for the most part, addressed in the revised manuscript. The manuscript in now suitable for publication in Nature Communications.

We thank the Reviewer for the positive review of our manuscript and for the constructive feedback that helped us improve this work.

Reviewer #4 (Remarks to the Author):

Zhou et al. proposed critical role of the phosphoinositide phosphatase Sac1 in regulating Golgi morphology and function through its impact on lipid composition, specifically PI(4)P and cholesterol levels. The study utilizes an auxin-inducible degradation system to rapidly deplete Sac1 from human cells, leading to a spike in PI(4)P and subsequent decrease in cholesterol levels within the trans-Golgi network (TGN). This shift results in TGN disintegration and impaired glycosylation processes, significantly affecting protein processing such as human chorionic gonadotropin (hCG) secretion. The findings highlight the Golgi V-ATPase as a key effector controlled by lipid exchanges driven by Sac1 activity, revealing insights into how lipid composition influences organelle integrity and cellular functions. The manuscript presents a compelling investigation into the role of Sac1 in regulating Golgi morphology and function through lipid composition, specifically focusing on PI(4)P and cholesterol dynamics. The use of an auxin-inducible degradation (AID) system to study acute effects following Sac1 depletion is innovative and allows for clear temporal analysis. The findings are significant in elucidating how lipid exchanges influence V-ATPase assembly and Golgi integrity. However, there are several areas where clarity could be improved, as well as opportunities to strengthen the discussion regarding broader implications as follow.

1. It would be good for reader with broader area of expertise if you could provide more context about previous studies about Sac1 functionality beyond its role as a phosphatase. Moreover, it would also be beneficial to include a brief overview of how dysregulation in these pathways may contribute to diseases associated with glycosylation defects.

Thank you for this remark. To our knowledge, a phosphatase independent function of Sac1 has not been reported. Multiple studies investigated the importance of the catalytic activity of Sac1 using a catalytic inactive Sac1 mutant (C389S mutation). Expression of this mutant in human cells or in yeast did not rescue the observed phenotypes of Sac1 KO, indicating that these phenotypes are dependent on Sac1 phosphatase activity (PMID: 34320354, PMID: 29461204, PMID: 14527956). We have now included this information (**line 121**). Similarly, in our hands, expression of a Sac1 phosphatase dead mutant (Sac1-C389S), could not rescue the selective degradation of B4GALT1 (**Fig. 2e, f**). Overall, it appears that there is currently no known phosphatase independent function of Sac1, including our present data.

Indeed, dysregulation of subcellular PI(4)P levels is linked e.g. to neurodegenerative disease and cancer (PMID: 33841102, PMID: 27178239, PMID: 24706697). We have now included this information in the Introduction (**line 34 – 35**). We have also added that previous literature has shown that Sac1 depletion can cause mislocalization of glycosylation enzymes, leading to defects in glycosylation (**line 44 – 45**).

2. In my opinion, to present the quantitative data from proteomics, consider using graphical representations alongside tables for easier interpretation.

We thank the Reviewer for this excellent suggestion. To present the quantitative data from proteomics, we show the GO analysis of the downregulated proteins in **Fig. 2a**. We have now also included the GO analysis of the upregulated proteins in **Suppl. Fig. 2b, line 106 - 107**. In addition to the data shown as a volcano plot in **Suppl. Fig. 2a**, we have now generated an interactive volcano plot where you can search and browse over the whole dataset and more easily identify the proteins (**Extended data 3, line 1417 - 1422**). Finally, in **Extended data 2**, there are graphical representations of the normalization, variation and correlation analysis of the proteomics data. We have now added a more extensive description in the legend belonging to Extended data file 2, to make it easier to find these data (**line 1410-1415**).

3. Strengthen discussions around observed changes in other phosphoinositide besides PI(4)P, even if they didn't show significant changes; any insights might foster deeper understanding.

As requested, we have now strengthened the discussion about the observed changes of the different PIP species. We added that the subcellular localization of different PIP species is important for organelle specific membrane properties and that dysregulation of the level or localization of PIP species can lead to multiple cellular defects in for instance migration, autophagy, endocytosis and signaling pathways. We also emphasized that the broad PIP imbalance likely affects multiple endomembranes upon long-term depletion of Sac1 (**line 388 – 392**).

4. Provide a more robust connection between laboratory findings and clinical implications; this

will enrich readers' understanding regarding how these molecular interactions translate into developmental processes or disease states.

This is an important (and broad) point. To our current knowledge, while the gnomAD analysis shows that the SACM1L gene has a high probability of being loss of function intolerant, there is no human disease directly linked to SAC1 perturbations. Combined with the fact that KO of Sac1 in mice leads to pre-implantation lethality, this suggests that loss-of-function of Sac1 may not be viable in humans. However, upon acute degradation of Sac1 we observed Golgi deacidification, fragmentation and degradation as well as cargo glycosylation and secretion defects. As we could partially rescue these phenotypes by treatment with the acidifying agent EN6, these phenotypes are most probably due to the loss of Golgi V-ATPase assembly caused by the altered TGN lipid environment. Interestingly, mutations in the V₀a2 subunit of V-ATPase are linked to trafficking and glycosylation defects, leading to connective tissue disorders, e.g. wrinkly skin syndrome and cutis laxa. We have now added this information to the Discussion (**line 346 – 348**).

5. Discuss potential feedback mechanisms between lipids that could affect cellular homeostasis beyond just direct perturbations caused by Sac1 loss. This could invite further exploration from future studies.

In the discussion (**line 336 – 340**), we have now elaborated on the cholesterol feedback mechanism in the ER. Previous studies have shown that SCAP can interact with the Sac1-VAP-OSBP complex, suggesting a coregulation between transcriptional control of cholesterol homeostasis and PI(4)P/cholesterol exchange (PMID: 33156328). Additionally, we added that PI(4)P is a counter exchange molecule for multiple lipid transfer proteins (PMID: 34817532), so that Sac1 manipulations are likely to affect multiple other lipids over time.

6. Detail of the setup parameters such as force fields used are crucial aspects when discussing simulation reliability. Please clearly specify the reason of the force fields utilized for the lipid bilayers and protein complexes. In your case, you have used CHARMM36m force fields for both proteins and lipid bilayer which may be too generalized because there are several well-known force fields separately designed for protein and lipids.

We thank the Reviewer for a well-founded comment, as the choice of force field is a key issue in simulations. In general, for each type of a molecule there are force fields that have been developed specifically to optimize the properties of this molecule type. However, combining different force fields and their parameters should be avoided. This is because, for example, different force fields often use different definitions for force field descriptions (dihedral definitions, etc.) and have been developed in different environments (different water models, etc.). Combining force fields may therefore lead to unexpected artifacts.

Currently, the state-of-the-art is that the best generic force field is CHARMM36m (PMID: 27819658). It provides a very accurate description for both folded and non-folded (disordered) regions of protein structures, lipids, and (with other CHARMM parametrizations also) ions. Given this, it is known as the force field of choice when complex systems comprised of several molecule types are being simulated.

It is also worth mentioning that we (in the Vattulainen group) have developed force fields for both atomistic and coarse-grained molecular representations over the last 25 years, and we systematically validate the force fields we use. The CHARMM36m force field we have used successfully (meaning in agreement with experimental data), e.g., in several recent studies (PMID: 39266533, PMID: 39043703, PMID: 38457493, PMID: 38252473, PMID: 37963916).

7. The MD simulations revealed that variations in lipid composition significantly affect the conformational states of the V_0 region of the Golgi V-ATPase. Specifically, higher levels of PI(4)P led to conformations that disrupt V_0 - V_1 complex assembly by destabilizing critical interactions between subunits. Conversely, introducing cholesterol into PI(4)P-rich membranes tend to promoted assembly of membrane proteins like V-ATPase by stabilizing certain conformational states conducive to binding with the intermembrane region. This underscores cholesterol's role as an essential modulator in maintaining protein complexes via lipid environment regulation. Therefore, the author should discuss more detail about their findings by provide a more quantitative comparison of how varying lipid compositions impacted V_0 region conformations.

We thank the Reviewer for these remarks. To provide a more quantitative comparison of how the lipid compositions affect the V_0 region conformations we have included the following additional metrics (text: **line 232 – 234, 243 – 245, 253 – 272, 369 – 371** and display items as detailed below):

First, we measured the number of interactions between the V_{0a2} head and PI(4)P molecules in the membrane in the different clusters. These interactions have been previously described (PMID: 28720663). Our data show that cluster 2 contains the most interactions between the a_2 head and PI(4)P, indicating that the arm and head of the a_2 subunit are positioned close to the membrane, supporting a conformation promoting V_0 - V_1 assembly (**Fig. 5g**).

The second metric measures the number of salt bridge interactions that stabilize the structure between the V_{0d1} subunit and the V_{0a2} head for each lipid composition (**Fig. 5i**). The number of these interactions is at its lowest when the V_0 - V_1 ATPase assembly occurs, forming an active structure. Here too, the composition 10% PI(4)P and 20% cholesterol leads to the lowest number of interactions between the a_2 head and V_{0d1} subunit. This indicates that a membrane containing 10% PI(4)P and 20% cholesterol produces an assembly-promoting and active V_0 - V_1 ATPase.

The final metric aims to describe how the lipid composition affects the ability of the V_0 region to bind to the V_1 region of the V_0 - V_1 complex: it quantifies the number of clashes (i.e., pairs of atoms at most 2Å apart) that the V_{0a2} subunit has with the V_1 region of the 6WM2 cryo-EM structure (see **Fig. 5k**). The smaller the number of clashes, the more likely the V_0 - V_1 complex

assembly is. In practice, for every snapshot of the simulations, we fitted the simulated structure of the V_0 subunit on the 6WM2 crystallographic form (human V-ATPase including the V_1 subunit). We then calculated for each lipid composition the average number of clashes between the V_0a2 arm and the V_1 subunit (**Fig. 5k-m**). When these data are interpreted through the clusters identified in the Machine Learning analysis, the results are reflected in the number of clashes observed in each cluster (**Fig. 5l**). Cluster 2, which has the most significant contribution to the assembly-promoting conformation, has also the smallest number of clashes (**Fig. 5l**). Therefore, the smallest number of clashes – indicating the greatest likelihood for the assembly of the V_0 - V_1 complex – is observed in the lipid composition of 10% PI(4)P and 20% cholesterol (**Fig. 5m**).

Please note that the four clusters presented in the paper (each representing different conformations of the V_0 subunit) are present in all simulated lipid compositions, but the relative proportion of these clusters depends on the lipid composition. The most important of these clusters is cluster 2, which is compatible with the assembly of the V_0 - V_1 complex, allowing for the formation of an active structure. The results show that cluster 2 originates mainly from a lipid composition of 10% PI(4)P and 20% cholesterol.